# Synthetic augmentation of cancer cell line multi-omic datasets using unsupervised deep learning

Zhaoxiang Cai [1,5], Sofia Apolinário [2,3,5], Ana R. Baião[2,3], Clare Pacini [4], Miguel D. Sousa[2,3], Susana Vinga [2,3], Roger R. Reddel [1], Phillip J. Robinson [1], Mathew J. Garnett [4], Qing Zhong [1] ✉ & Emanuel Gonçalves [2,3] ✉

Integrating diverse types of biological data is essential for a holistic understanding of cancer biology, yet it remains challenging due to data heterogeneity, complexity, and sparsity. Addressing this, our study introduces an unsupervised deep learning model, MOSA (Multi-Omic Synthetic Augmentation), specifically designed to integrate and augment the Cancer Dependency Map (DepMap). Harnessing orthogonal multi-omic information, this model successfully generates molecular and phenotypic profiles, resulting in an increase of 32.7% in the number of multi-omic profiles and thereby generating a complete DepMap for 1523 cancer cell lines. The synthetically enhanced data increases statistical power, uncovering less studied mechanisms associated with drug resistance, and refines the identification of genetic associations and clustering of cancer cell lines. By applying SHapley Additive exPlanations (SHAP) for model interpretation, MOSA reveals multi-omic features essential for cell clustering and biomarker identification related to drug and gene dependencies. This understanding is crucial for developing much-needed effective strategies to prioritize cancer targets.

The growing molecular and phenotypic characterization of cancer cell lines makes them one of the most studied human cell models[1]. This ever-growing and rich multi-omic data continues to drive the identification of cancer genes and the discovery of therapeutic targets[2–4]. Although genomics has been a primary focus in the search for predictive biomarkers in cancer, recent functional genetic screens conducted by the Cancer Dependency Map (DepMap) consortium revealed that less than 20% of RNAi cancer dependencies could be explained by mutations and copy number alterations[5]. This highlights the importance of developing holistic machine learning models capable of vertically integrating orthogonal datasets. In this case, vertical integration involves not only genomics but also other types of omics data[6].

Despite recent successes of deep learning[7] multi-omics integration faces several limitations, most importantly high heterogeneity of different data types (e.g., discrete vs. continuous distributions), intrinsic technological limitations (e.g., missing values), and limited data availability (e.g., in this study, only 25.8% of the cancer cell lines have a complete set of all seven omic datasets under consideration)[8]. Unsupervised machine learning has been successful in multi-omics integration capturing patterns of data variation shared across different omics[9,10]. This approach highlighted cancer cellular states associated with epithelial-to-mesenchymal transition (EMT), a key process in drug resistance and metastasis[11]. Unsupervised deep learning based models can generate improved versions of input datasets by reconstructing missing measurements and correcting experimental error, and

[1]ProCan®, Children's Medical Research Institute, Faculty of Medicine and Health, The University of Sydney, Westmead, NSW, Australia. [2]INESC-ID, 1000-029 Lisboa, Portugal. [3]Instituto Superior Técnico (IST), Universidade de Lisboa, 1049-001 Lisboa, Portugal. [4]Wellcome Sanger Institute, Wellcome Genome Campus, Cambridge CB10 1SA, UK. [5]These authors contributed equally: Zhaoxiang Cai, Sofia Apolinário. ✉e-mail: qzhong@cmri.org.au; emanuel.v.goncalves@tecnico.ulisboa.pt

thereby augmenting downstream analysis[12,13]. Although linear dimensionality reduction models[10,14] have been designed for similar purposes, the application of deep generative models to large-scale multi-omic cancer cell models is lagging behind. This leaves a gap in the utilization of these non-linear approaches to augment datasets and perform statistical analysis to improve the characterization of cancer mechanisms, biomarkers and drug targets[5,15,16]. Deep learning models, such as variational autoencoders (VAE), provide more complex formulations of the underlying biological data. Moreover, VAEs have highly flexible designs that can handle data sparsity robustly and are easily extensible to incorporate different data types. In particular, methods based on VAE models have demonstrated significant success in the field of single-cell multi-omics integration and augmentation. However these methods often presuppose the presence of specific data types, such as count data from scRNA-seq and scATAC-seq, limiting their applicability across broader omic landscapes[17–21].

Here, we developed a Multi-Omic Synthetic Augmentation (MOSA) VAE model that integrates and synthetically augments multi-omic datasets from >1500 cancer cell lines of the DepMap. MOSA provides a generative unsupervised deep learning model for cancer discovery that utilizes SHapley Additive exPlanations (SHAP)[22] values for model explainability, facilitating the identification of underlying biological mechanisms and drug targets. In our study, we systematically evaluated and benchmarked MOSA, demonstrating its generative capacity across independent drug response and proteomic datasets and accurately recovering cancer tissue-of-origin clustering. Additionally, MOSA increased the statistical power to find genomic associations with CRISPR-Cas9 gene essentiality screens. Synthetically screened cancer cell lines revealed vulnerabilities consistent with genomic profiles, such as *FLI1-EWSR1* fusion dependency. With MOSA, we generated a complete multi-omic profile across all seven different omics, increasing by 32.7% the number of available screens.

## Results

### Unifying deep generative model for cancer multi-omics

Taking advantage of the DepMap project[5,6,23,24], we assembled seven different cancer cell line datasets, i.e., genomics[2,3], methylomics[25], transcriptomics[26], proteomics[11], metabolomics[27], drug response[2,25,28,29], and CRISPR-Cas9 gene essentiality[4,30] (Fig. 1a). This comprises a total of 1523 cancer cell lines for which at least two datasets were available (Supplementary Data 1). We designed MOSA tailored to the cancer cell lines multi-omic datasets, performed robust data augmentation, and provided model explanations for biomarker discovery (Fig. 1b, see Methods).

First, following a late integration[31] approach, we trained a separate encoder for each dataset to derive latent embeddings specific to each omic layer. These embeddings were then concatenated and further reduced to formulate a joint multi-omic latent representation (Fig. 1c, Supplementary Data 2). Here, a latent representation is a learned, abstracted feature set (embeddings) within the hidden layers of the neural network that encapsulates the major information from the input data. Compared to a multi-omic linear dimensionality reduction method, MOFA[10,14], and another VAE-based method MOVE[32], our model provides better separation of cell lines by tissue in the multi-omic latent space (Fig. 1c, Supplementary Fig. 1).

Second, genomics presents a unique challenge due to the sparsity and qualitative nature of its data. To address this, we use only cancer driver events and split genomics into copy number alterations and mutations. While copy number events are integrated as ordinal data through a separate encoder/decoder akin to other omics, mutations are integrated as binary conditionals to each encoder (Fig. 1b, see Methods). The rationale is that genetic backgrounds influence cellular profiles and phenotypes, thereby conditioning other omic layers. The conditional matrix contains genetic alterations in cancer driver genes (including gene fusions), cell line tissue of origin, cell line growth rate

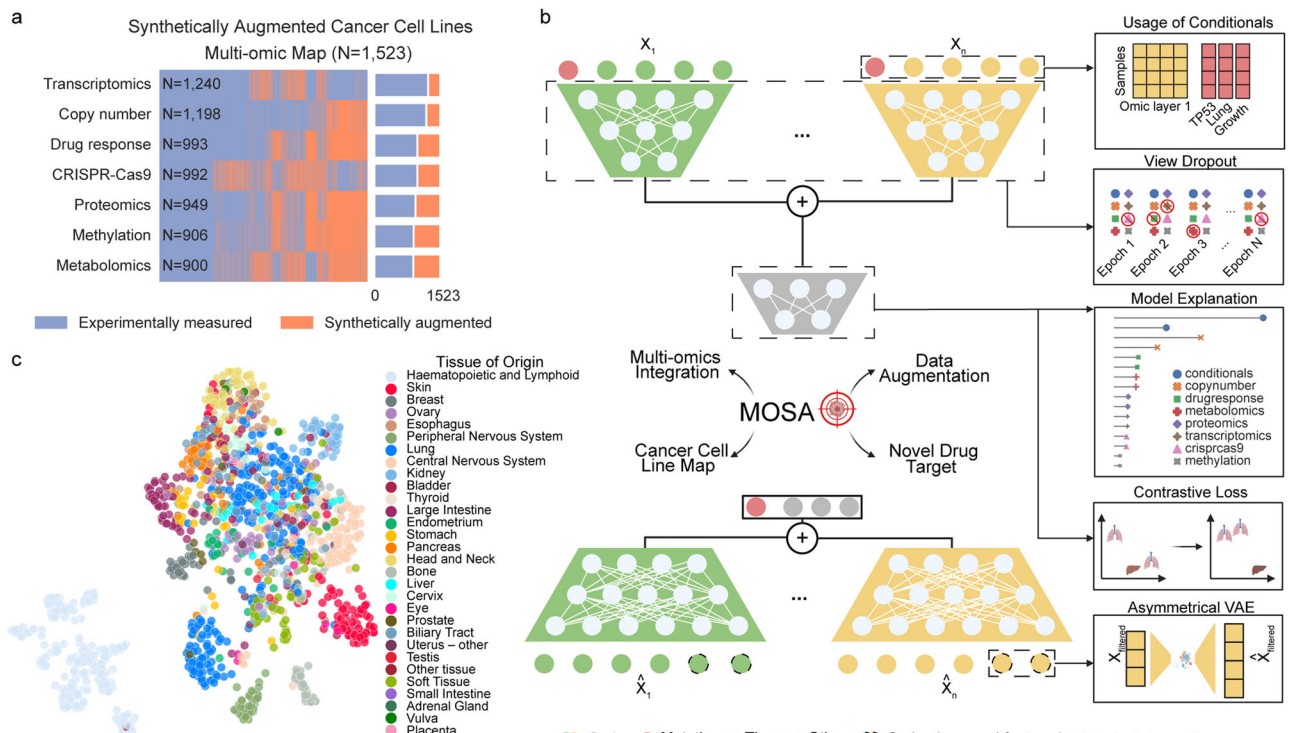

**Fig. 1 | Cancer multi-omics integration with MOSA. a** Cancer cell line multi-omic datasets across the 1523 cancer cell lines. Purple represents measured screens, while orange represents gaps, i.e., missing screens, which were synthetically generated with MOSA. **b** Schematic of the autoencoder, MOSA, where encoders are represented at the top and decoders at the bottom. For simplicity, the integration of only two datasets is represented. Highlighted designs of MOSA are illustrated on the right. Created in BioRender. Cai, Z. (2023) BioRender.com/m96b457. **c** Dimensionality reduction visualized using Uniform Manifold Approximation and Projection (UMAP) representation of the trained MOSA joint latent space, where each dot represents a cancer cell line colored according to its tissue of origin.

measurements, and microsatellite instability information (MSI high), totaling 237 conditional variables (Supplementary Data 3). This conditional matrix is further concatenated to the learned multi-omic joint latent space that works as input for the decoders. Hence, the genetic background and cellular information are crucial for generating latent representations and reconstructing each omic dataset.

Third, compared with similar models for single-cell data[12,21,33], the limited number of samples and heterogeneity of the omics available in the DepMap pose significant challenges to training a generalizable model for cancer cell lines. To reduce model complexity, MOSA only considers the most variable features as input for the encoders, while all features are reconstructed by the decoders for synthetic data generation, resulting in an asymmetrical design of VAE (Fig. 1b, Supplementary Fig. 2a, b, Supplementary Data 4). This unique design of MOSA allows us to discard low informative features, such as genes with constant expression and non-essential genes across all cancer cell lines. This reduces the number of trainable parameters by 39.2% while maintaining low reconstruction error.

Fourth, the diverse size of the omic multi-omic datasets may lead to some datasets dominating during training, diminishing the model's generalizability and explainability. We develop a whole omic (view) dropout layer, which masks a complete omic layer based on a hyperparameter. This provides a significant improvement in the model's generalization, providing better reconstructions for cancer cell lines by specific omics (see Methods, Fig. 1b). We then perform a multi-omic model explanation by calculating SHAP values[22] for all omic input features to assess their importance for the latent space integration and the reconstruction of omic features (see Methods). This provides a systematic resource to explore potential nonlinear cancer genotype-phenotype associations.

Taken together, MOSA provides an unsupervised model that integrates all cancer cell line omics simultaneously. Using a 10-fold cross-validation strategy, MOSA's reconstructed hold-out folds for CRISPR-Cas9 and drug responses were robustly correlated with the original data (mean feature Pearson's r of 0.35 and 0.65, respectively) (Fig. 2a, Supplementary Fig. 3, Supplementary Data 5). MOSA performed better compared to a similar systematic supervised analysis designed to predict each CRISPR-Cas9 gene dependency either using core-omics (e.g., genomics, transcriptomics), only genomic, or only functionally related genes (mean feature best Pearson's r = 0.25)[34].

## Evaluation of multi-omics synthetic data generation

A significant advantage of multi-omic vertical integration and unsupervised deep-generative models is their ability to synthetically generate datasets that are missing in specific samples, such as reconstructing a dataset that is entirely absent for certain cell lines. This is particularly crucial given the pervasive dataset gaps, even in well-characterized models such as cancer cell lines (Fig. 1a). Multi-omic profiling is both costly and labor-intensive, thus data-driven generative models are key to prioritizing the design of the most informative experiments. However, benchmarking generative models is challenging as it requires independent, ideally large-scale, datasets to validate the model's predictions. We initially tested 16 multi-omics integration methods (Supplementary Data 6, see Methods), but due to constraints such as the number of omics supported, type of data distribution, and limitations in design and implementation, we were narrowed to three state-of-the-art methods: MOFA[10,14], MOVE[32], and mixOmics[35,36]. These methods, encompassing linear, VAE-based, and correlation analysis approaches, were capable of integrating all seven omics datasets considered here. We have delineated a series of benchmarks over the following sections into increasing model complexity.

MOSA reconstructs the input data matrices by leveraging the multi-omic latent space learned from the original data. Data reconstruction generates complete omic matrices, thus handling both missing values (partial dataset augmentation), and more importantly,

reconstructing whole-omics (full dataset augmentation) through vertical integration (at least two omics are required for a cell line to be considered in this study). For partial dataset augmentation, MOSA imputes incomplete features, e.g., measurements for certain proteins are sparse due to technical limitations commonly found in mass-spectrometry-based proteomics data[11,37]. A recent and independent drug response dataset, that is completely absent during model training, was accurately reconstructed (IC50s, Pearson's r = 0.87, n = 32,659) (Fig. 2b), outperforming MOFA[10,14], MOVE[32] and naive mean imputation (Fig. 2c–e). Pronounced discrepancies between MOSA's reconstruction and the original datasets revealed likely inaccurate experimental measurements. For example, the response to the MEK1/2 inhibitor trametinib was not consistent with replicate measurements and drugs with the same canonical target in the same cell line (Supplementary Fig. 4a). Such discrepancies also spotlighted drugs (e.g., venetoclax) or classes of drugs (e.g., antiapoptotic inhibitors) for which no effective molecular biomarkers are available (Supplementary Fig. 4b), underlining the challenge of devising reliable predictive models for their response. Additionally, proteomics is riddled with missing values (Supplementary Fig. 5a), affecting more predominantly lowly abundant proteins. MOSA augmented the proteomics data by filling approximately 32% of the original matrix using information from all omics, while preserving sample correlations with an independent proteomic dataset (CCLE[38]) (Supplementary Fig. 5b, Supplementary Data 7). Notably, MOSA effectively reconstructed the protein profiles of *SMAD4* in cell lines characterized by *SMAD4* gene deletions, which are typically associated with low *SMAD4* gene expression and protein abundance (Supplementary Fig. 5c). The MOSA-augmented proteomic matrix preserves the ability to identify protein interactions through protein pairwise correlations[39] (Supplementary Fig. 5d). In contrast to the original matrix that has missing values, MOSA's augmented protein matrix is complete and directly usable for downstream analysis, such as generalized linear models[40], which improved the recall of protein complex interactions (Supplementary Fig. 5d).

Subsequently, full dataset augmentation was assessed. Synthetic proteomic data generated by MOSA for cancer cell lines lacking proteomic measurements showed correlations with independent proteomic measurements comparable to those of cell lines that had actual proteomic data (Fig. 3a). For drug response, reconstructions of 107 overlapping drugs correlated robustly with measurements in an independent dataset (CTD2[41,42]) (Fig. 3b). Lastly, we performed a similar analysis using independently processed transcriptomics, which included data for 272 cancer cell lines that did not have transcriptomics data during the training of MOSA[26]. MOSA's transcriptomic reconstructions were strongly correlated with real data even for cell lines with no transcriptomics data for training (mean pearson's r = 0.90) (Supplementary Fig. 5e). Crucially, this shows the capacity of MOSA as a generative model for synthetic cancer cell line multi-omic and phenotypic screening.

We evaluated downstream analysis by comparing the original data matrices with the augmented ones. MOSA increased by 34.9% the number of CRISPR-Cas9 cell line screens, and the augmented dataset improved the statistical power to find genetic associations (Fig. 3c, Supplementary Data 8). Gene essentiality specificity (Fisher's skewness test), which can be used to identify selective cancer vulnerabilities, showed a moderate positive correlation (Pearson's r = 0.52) between the synthetic CRISPR-Cas9 screened cell lines and the previously available screens (Fig. 3d). Nonetheless, this correlation is likely underestimated due to the presence of potential outlier non-essential genes. MOSA accurately reconstructed gene dependencies, for example, *BRAF* dependency in *BRAF* gain-of-function mutant cancer cell lines (Fig. 3e), and *FLI1* dependency in cell lines harboring an *FLI1-EWSR1* fusion gene (Fig. 3f).

Lastly, we aimed to assess the advantages of developing a method capable of natively integrating more than two omics. Specifically, we

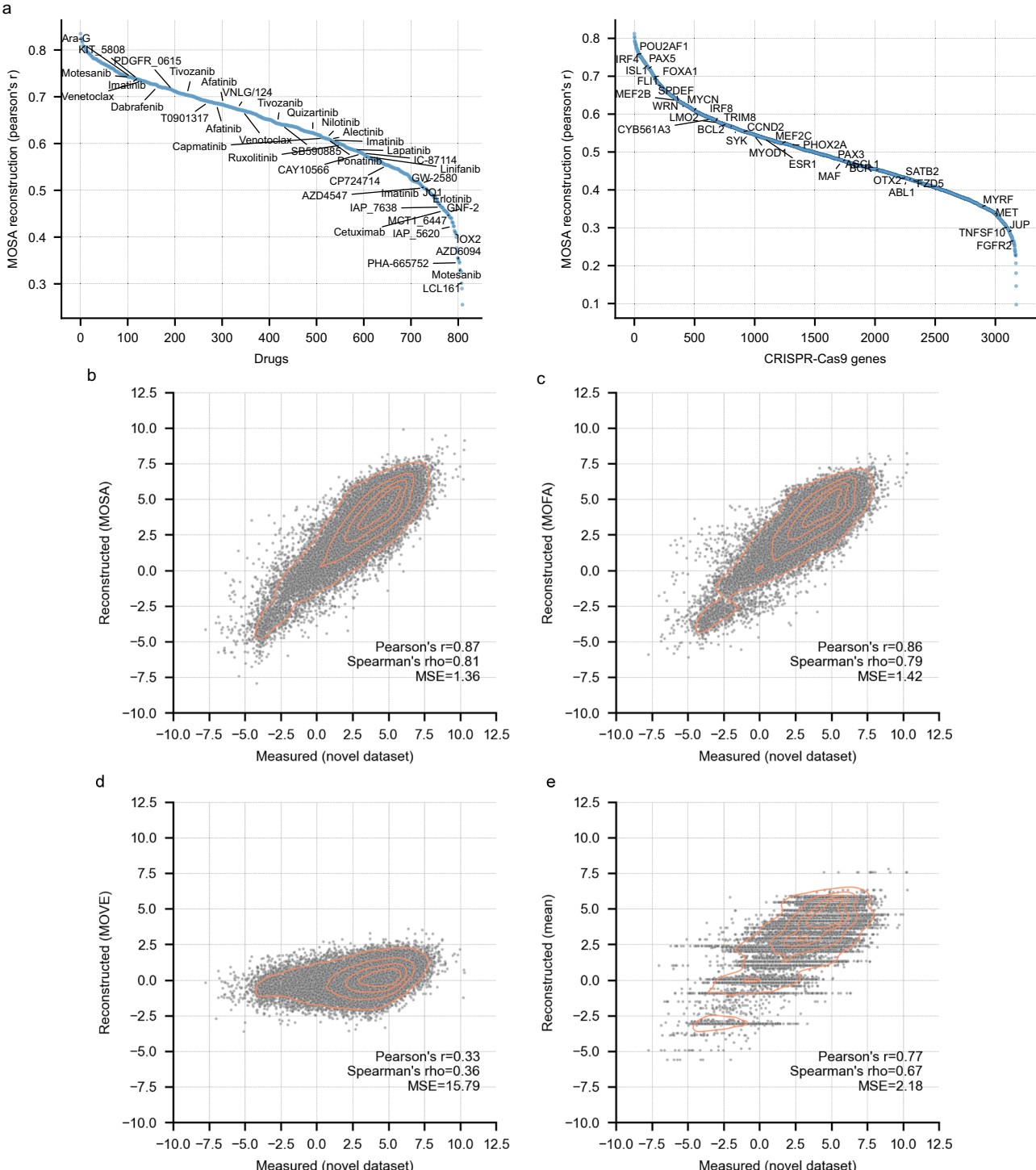

**Fig. 2 | MOSA reconstruction of drug response and CRISPR-Cas9 datasets.**
**a** MOSA reconstruction quality measured using a 10-fold cross-validation. After reconstructing all test folds, they are concatenated and the reconstruction quality score is calculated as the Pearson's r between the reconstructed and actual measured values. Features ranked by their reconstruction quality are shown for the drug response (left) and the CRISPR-Cas9 (right) datasets. Duplicated drug names represent replicated screens for the same drug. Representative examples of strongly selective CRISPR-Cas9 and drug responses are labeled. **b** MOSA's partial dataset augmentation (missing value imputation) of drug IC50s compared to recent independent drug response screens. **c**–**e**, similar to **b**, using MOFA, MOVE and mean imputed values, respectively.

focused on transcriptomics and drug response datasets, which represent molecular and phenotypic datasets, respectively. These are also commonly utilized in multi-omics integration and are among the most informative omic types for our benchmarks. From the list of methods we evaluated, we considered iClusterPlus[43], JAMIE[18], scVAEIT[44], and moCluster[45] (Supplementary Data 6, see Methods).

MOSA provided a better reconstruction of transcriptomics and drug response data (Supplementary Fig. 6a, b). Particularly, adding more omics to MOSA provided a significant improvement over existing methods, supporting the utility of using holistic multi-omics models. Furthermore, MOSA consistently outperformed the other methods in tissue of origin clustering (Supplementary Fig. 6c). Considering only

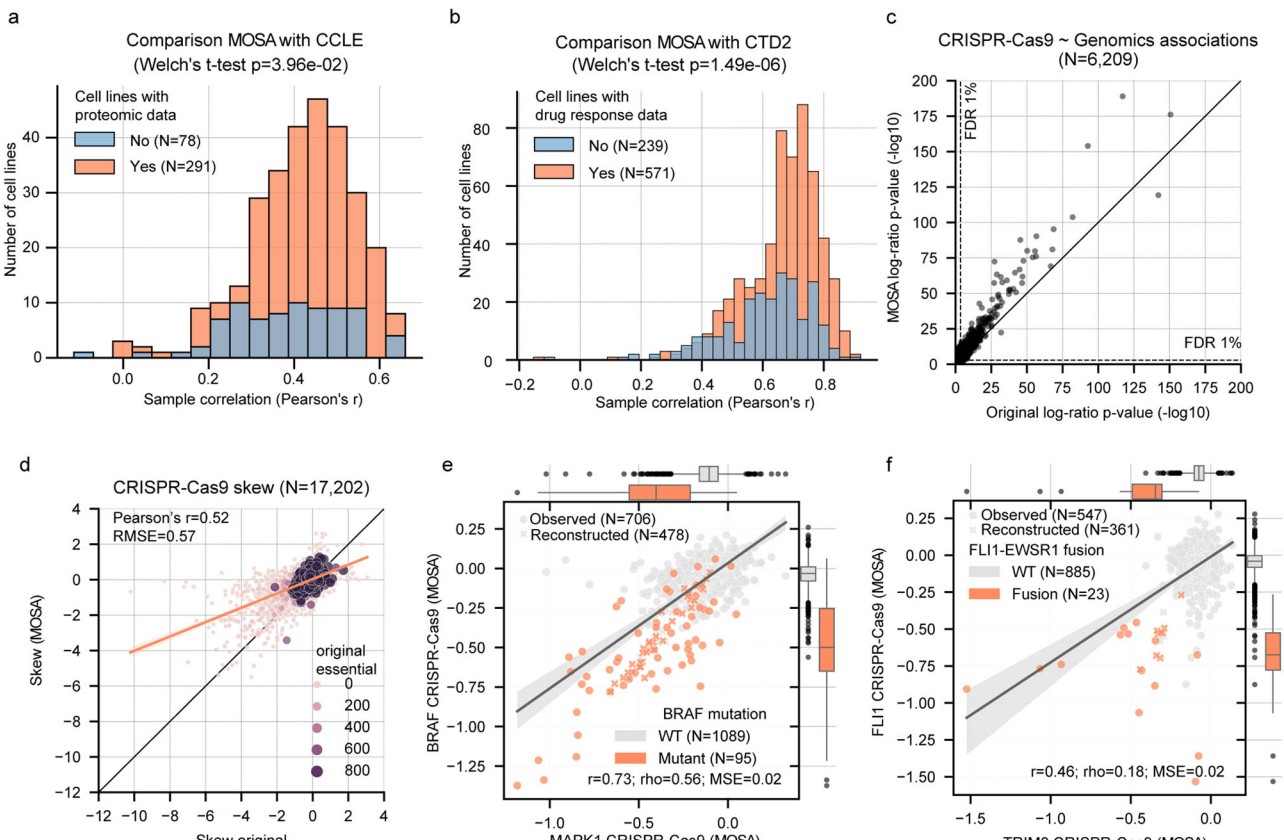

**Fig. 3 | Multi-omics benchmark of MOSA. a** Distribution of proteomics cancer cell lines correlation with an independent dataset (CCLE[38]) grouped by whether the cancer cell line had proteomic data for the model training (orange, *n* = 291) versus cell lines without any proteomics prior (light blue, *n* = 78). **b** Distribution of cancer cell line correlations (Pearson's r) between an independent drug response dataset (CTD2[41,42]) and the MOSA reconstructed dataset, grouped by whether the cancer cell line had prior availability of drug response in the datasets for the model training (orange, *n* = 571) versus cell lines without drug response data (light blue, *n* = 239). **c** One-sided log-ratio test p-value of genetic associations with CRISPR-Cas9 gene essentiality with the original dataset (x-axis) and the augmented MOSA dataset (y-axis). False discovery rate (FDR) correction is applied using the Benjamini-Hochberg method to adjust for multiple comparisons. **d** Fisher skew test per gene across the original CRISPR-Cas9 dataset (x-axis) and the MOSA augmented dataset (y-axis). Dot size represents the number of cell lines that have the gene as essential (scaled log2 fold-change < −0.5) in the original dataset. **e** Correlation between *BRAF* and *MAPK1* CRISPR-Cas9 gene essentialities using both previous measured (Observed) and the synthetically reconstructed (Reconstructed). Gene essentiality scores are represented using copy-number corrected[78] log2 fold-changes scaled by the median of common essential (score = −1) and non-essential (score = 0) genes[30]. Gene essentialities are also grouped according to the presence or absence of a *BRAF* mutation, mostly V600E gain-of-function mutations. **f** CRISPR-Cas9 gene essentiality association with *FLI1-EWSR1* fusion. Confidence intervals of 95% are displayed for the regression lines in panels **d**, **e**, and **f**. Box-and-whisker plots show 1.5× interquartile ranges, centers indicate medians in panels **e** and **f**.

transcriptomics and drug response resulted in the best tissue of origin clustering, reflecting the strong structuring of these omics by tissue of origin[46]. In contrast, other omics such as proteomics and metabolomics are more loosely structured by tissue[11,47]. Consequently, including omics that are less strongly structured by tissue will naturally result in looser tissue clustering.

Taken together, these diverse examples demonstrate MOSA's ability to perform both partial and full dataset augmentation validated using various independent datasets and from different laboratories. The generation of large-scale multi-omic datasets is both time and resource-intensive, thereby positioning MOSA as a valuable tool for in silico testing and prioritization of drug targets for experimental validation.

## Model interpretation reveals cancer cell states

To prioritize the most promising targets, a model needs to be explainable beyond producing reliable predictions. Hence, we used the SHAP[22] algorithm to calculate the feature importance, defined as the amount of contribution of each feature to the latent space (Fig. 1b, Supplementary Data 9, see Methods). When grouping features by their corresponding omic datasets, we observed that metabolomics, drug response, and

copy number alterations exhibited the highest average feature importance (Supplementary Fig. 7a). Regarding conditional features, although their average feature importance was modest, certain key features, such as *TP53* mutation, growth rate, and tissue of hematopoietic and lymphoid origin emerged as highly significant, even when compared with other omic datasets (Fig. 4a, Supplementary Data 9). This underscores the importance of incorporating conditional variables into the model. Features ranked in the top five from each omic dataset also validated the capacity of our approach to recover well-established molecular processes associated with cancer (Fig. 4a), for example, *CDKN2A* copy number alterations, as well as sensitivity to the SRC family inhibitor, dasatinib. Interestingly, other less obvious features that were highly ranked shed light on previously less explored biological mechanisms. One specific example is the metabolite, 1-methylnicotinamide involved in the nicotinate and nicotinamide metabolism, which was calculated to be the most important feature in the metabolomics towards the multi-omics latent representation (Fig. 4a). We observed a strong relation between increased 1-methylnicotinamide intracellular abundance and the overexpression of Nicotinamide N-Methyltransferase (NNMT) enzyme, which catalyzes the production of this metabolite (Supplementary Fig. 7b). We also observed an association between

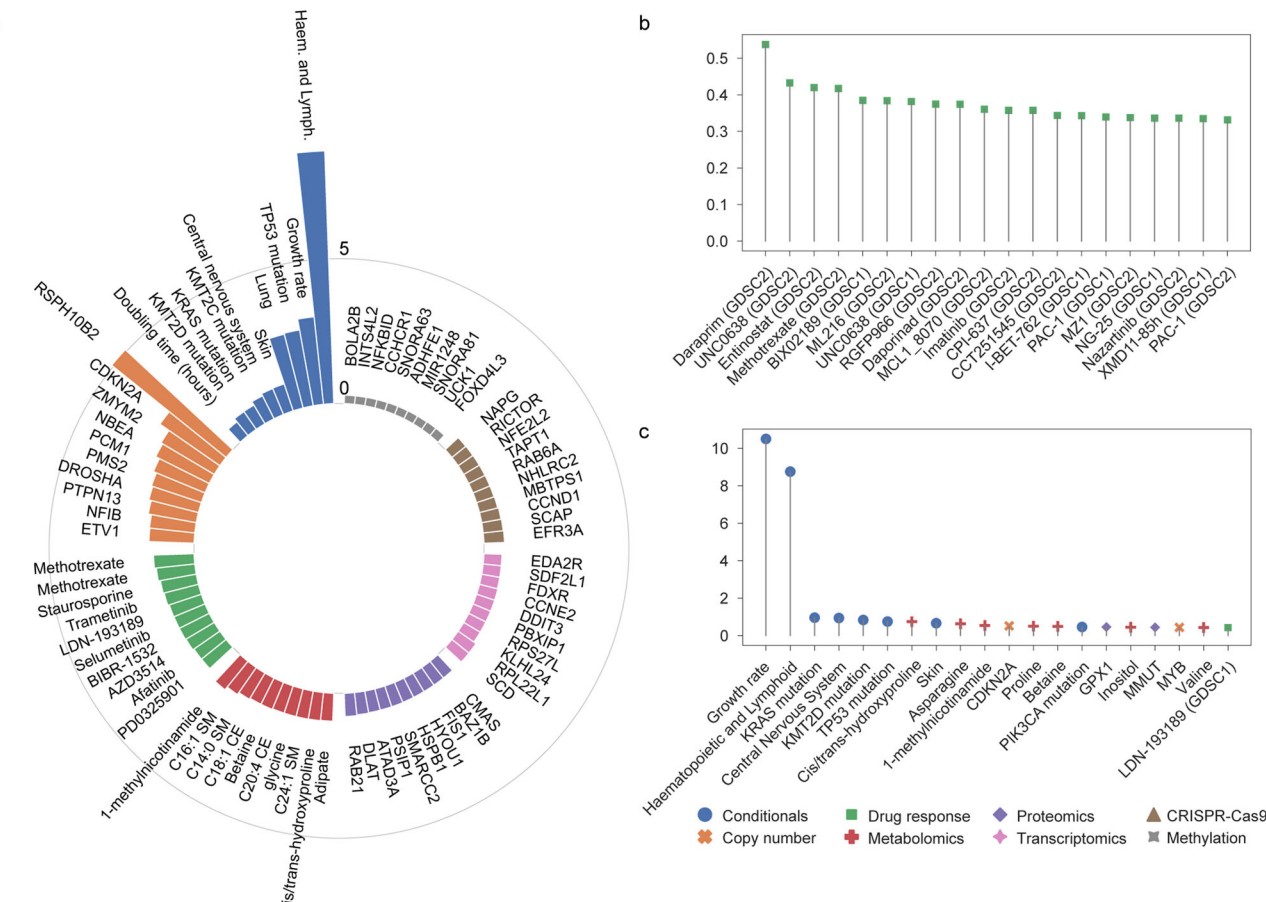

**Fig. 4 | SHapley Additive exPlanations (SHAP) model explanation of MOSA.** **a** Top features from each omic layer that contribute the most to the multi-omic latent space. **b** Top drugs that have the highest feature importance from metabolite 1-methylnicotinamide. **c** Top features that contribute the most to the reconstruction of the drug response of Daraprim (Pyrimethamine).

1-methylnicotinamide and the EMT state of cancer cell lines, as corroborated by the expression of *VIM* and *CDH1*[11] (Supplementary Fig. 7c, d). This confirms a recent single-cell study's finding that the PC-9 non-small cell lung carcinoma line, which harbors an activating *EGFR* mutation, develops a cellular state resistant to EGFR inhibitors through expression of EMT markers as well as accumulation of 1-methylnicotinamide[48]. Additionally, 1-methylnicotinamide was observed with a significant increase during the early stages of EMT in the A549 cell line, and this increase was associated with changes in glycolytic metabolites and histone post-translational modifications, indicating a link between 1-methylnicotinamide and epigenetic modifications during EMT[49]. While further experimental validation is necessary, this could pave the way for the identification of cancer cellular states underlying drug resistance.

To delve deeper, we subsequently used the SHAP algorithm to calculate the feature importance specifically for the reconstruction of drug response, thereby facilitating the discovery the most promising biomarkers (see Methods). As expected, the drug response features themselves were the most important features on average (Supplementary Fig. 8a, Supplementary Data 10). Notably, the conditionals emerged as the second most important omics, reflecting the critical role of tissue of origins, mutations, and growth rate in influencing drug responses (Supplementary Fig. 8a). Centering on the metabolite 1-methylnicotinamide, drugs known to be EMT-related were ranked as the top drugs showing high feature importance from 1-methylnicotinamide (Fig. 4b). All the top five drugs, except Daraprim (Pyrimethamine) which was not included in the

dataset as an anti-cancer drug, were found to be related to EMT in recent studies. Specifically, UNC0638[50], Entinostat[51], and BIX02189[52] suppress EMT, while methotrexate[53] shows the ability to induce EMT. This finding suggests that the top-ranked drug Daraprim may also harbor a close relation to EMT, presenting a potential avenue for repurposing in cancer treatment. Other EMT-related features such as GPX1 protein intensity[54] also ranked as top features for Daraprim, indicating the potential to utilize other features in the list for the discovery of the most promising biomarkers for drug response (Fig. 4c). Among the other top features for the top drugs, *KRAS* and *KMT2D* were consistently identified as being of high importance, and both of these genes have been implicated in EMT[55,56] (Supplementary Fig. 8b–f). Lastly, we utilized an external metabolomic dataset[47] to validate the drugs associated with 1-methylnicotinamide by SHAP values. Although the abundance of 1-methylnicotinamide was not directly measured in their study, we analyzed the drugs linked to nicotinate and nicotinamide metabolism, where 1-methylnicotinamide is a direct product of nicotinamide methylation. Several highlighted drugs identified by SHAP values, including Daraprim, UNC0638, Entinostat (MS-275), and PAC-1, were also ranked highly as either resistant or sensitive drugs (Supplementary Fig. 9).

Taken together, our findings suggest a broad association of 1-methylnicotinamide and EMT across hundreds of cancer cell lines with a potential role in drug resistance. While further assessment is needed to substantiate this, more generally, it unveils the possibility of using MOSA as a holistic model that integrates molecular and

phenotypic data of cancer cells to investigate cancer cell states, drug resistance and their underlying mechanisms.

## Discussion

The application of deep generative models, including MOSA, in cancer research is promising but comes with limitations, mainly related to the restricted sample size which impaired exploring more complex VAE designs, and more complex designs led to worse dataset reconstructions. While the overall reconstruction of the datasets was robust, there are examples where it could be improved, particularly for proteomics where intrinsic data sparseness makes it more challenging for the model to train successfully. Thus, the addition of more characterized cancer models will likely allow us to train better models and reduce reconstruction error. Future efforts should leverage multi-omic resources from cancer patients and derived models, such as organoids and patient derived xenografts (PDXs), to enhance training and explore transfer learning opportunities. In addition to tabular omic data, VAEs have demonstrated great success integrating image and text-based data[57,58], and MOSA can be further enhanced to integrate these types of data and enable multi-modal data augmentation. We also aim to address the complex challenge of data missing not at random (MNAR), a scenario commonly encountered in omics datasets, by adapting VAE architectures to more accurately identify and handle MNAR scenarios[59-61]. SHAP analysis offers an explanation for deep learning models, however, there are still some obstacles in verifying the biological significance of certain highlighted features. These challenges could be associated with the inherent limitations of SHAP and Shapley values[62], thus additional research is required to ascertain the importance of these emphasized features. Furthermore, while this has provided strong initial support for the EMT-related associations, further experimental work is necessary to validate and confirm these findings across different cancer cell models.

In summary, MOSA augmented the multi-omic profiles of 1523 cancer cell lines by robustly filling in gaps in the existing experimental screens. Deep learning-based synthetic data generation can augment experimental screens by facilitating the creation of realistic datasets to guide experimental design and accelerate the validation of the most promising targets. Looking ahead, this model is readily adaptable to integrate other types of data modalities, such as imaging, further enabling the discovery of molecular/phenotype associations.

## Methods

### Cancer cell line multi-omic data collection

The aim was to assemble the most up-to-date and comprehensive molecular, phenotypic and cancer cell line sample information. All datasets were downloaded from the DepMap (https://depmap.org/), and the CellModelPassports (https://cellmodelpassports.sanger.ac.uk/)[23] portals, with the exception of the metabolomics data which were taken directly from the original publication supplementary materials[27]. For reproducibility, all data used in this study are provided in a figshare repository (see Code and data availability).

We integrated genomics[2,63], transcriptomics[26], methylomics[25], proteomics[11], metabolomics[27], drug response[25,28,29], and CRISPR-Cas9 gene essentiality[4,64]. This comprised a total of 1523 cancer cell lines with at least two datasets available for each cell line. All datasets have been previously processed, normalized/scaled, and batch corrected in each of their individual publications addressing technical and design aspects important to each dataset (e.g., integration of CRISPR-Cas9 screens across different laboratories[65], driver mutations and copy number alterations, and gene expression samples from different datasets[26]).

### Cancer cell line validation datasets

Three independent datasets were used in this study for validation, i.e. they were not used for model training. The first dataset presents the CCLE proteomic characterization of 375 cancer lines[38], of which 291 comprise the proteomic dataset[11] used for training. The second dataset represents recent drug response screens with the same platform as the drug screens used for training[25,28,29] that were obtained from the Genomics of Drug Sensitivity in Cancer (GDSC) portal (https://www.cancerrxgene.org/)[66] comprising a total of 32,659 IC50s measured across 313 unique drugs and 781 overlapping cancer cell lines. The third dataset is an independent drug response dataset (CTD2)[41,42], comprising a total of 545 drugs and 887 cancer cell lines, for which 106 and 575, respectively, overlap with the drug response data used for training[25,28,29].

### Data preprocessing

A total of seven datasets were considered: copy number ($n = 777$ features); methylome ($n = 14,608$); transcriptome ($n = 15,278$); proteome ($n = 4922$); metabolome ($n = 225$); drug response ($n = 810$); and CRISPR-Cas9 gene essentiality ($n = 17,931$). A total of 1523 cancer cell lines were profiled with each cell line having at least two of these datasets.

For CRISPR-Cas9 gene essentiality, transcriptomic and methylomic feature reduction was performed to exclude lowly variable features. For gene essentiality, samples were scaled using essential and non-essential genes making their median per sample -1 and 0, respectively. Never essential genes were discarded, i.e., genes that do not have an essentiality profile lower than 50% of the median log2 fold-change of essential genes in at least one cell line were removed. For transcriptomics and methylomics, a standard deviation filter was applied. By taking the standard deviation of all genes across samples, a Gaussian mixture model (k = 2) was fitted, identifying lowly variable genes and the rest. A standard deviation threshold was defined as the rightmost intercept of the two Gaussian distributions (Supplementary Fig. 2a), and any gene with a standard deviation lower than that was discarded. Moreover, for the proteomic, drug response, metabolomic and CRISPR-Cas9 datasets, any feature with a missing rate higher than 85% was discarded. All datasets were standardized by z-score, except copy number. Missing values were replaced with 0 and their position in the original dataset was stored for use in the model (e.g., to exclude them from the loss functions). In addition to these seven datasets, driver gene mutations, fusion genes, microsatellite instability, growth rate, cancer and tissue type information were concatenated into a single matrix to be used as labels of the cancer cell lines.

### Multi-omics synthetic augmentation (MOSA)

MOSA is a conditional multi-view variational autoencoder implemented using PyTorch (v2.0)[67]. In the next section, we describe MOSA's architecture, use of conditionals, dropout layer and SHAP explainability analysis.

**Architecture.** MOSA follows a traditional design of conditional VAEs (Fig. 1b). For each of the seven datasets (views), an encoder is trained with multiple fully connected layers, which are all proportional to the number of input features of the dataset plus the number of labels (concatenated conditionals). First, joint fully connected layers take as input each dataset and reduce them to a fixed number of joint latent dimensions. Different techniques were tested to integrate the omic-specific latent dimensions (e.g., product of experts), but concatenation obtained the smallest reconstruction loss. The multi-omics joint latent dimensions are further reduced to a specified number of latent dimensions (hyperparameter). Then, the joint layer outputs two layers representing Gaussian distribution mean and variance. These are important for the regularization of the latent space and are used to sample the latent dimensions. Finally, the latent dimensions (z) are concatenated with the conditionals and provided to the decoders of each dataset. The decoders have a similar but inverse architecture to the encoders.

**Conditionals**. We introduced a conditional architecture to enhance the model's reconstruction performance and biological relevance. Conditionals ($n = 237$) include key biological features, such as cancer driver mutations, tissue types, gene fusions, MSI status, and cell line growth rate. These were used in two stages in model architecture: 1) concatenated to each omic layer prior to encoding; 2) concatenated to the multi-omic joint latent representation before decoding. The conditional concatenation serves two crucial purposes: it contextualizes the input data within specific cellular or genetic backgrounds, and it allows the decoder to generate condition-specific reconstructions of the data. The inclusion of conditionals offered several advantages. First, it ensured that the model was not merely capturing patterns within individual omic layers in isolation. Instead, complex interactions among multi-omic data and genomic and physiological variables were accounted for, facilitating a more holistic understanding of the underlying biological processes and phenomena. Second, by embedding these conditionals into the decoder, the model can generate data reconstructions contextualized to specific cell line conditions.

**View dropout layer**. A special dropout strategy, namely the view dropout layer, was included in MOSA to both improve the model's predictive power and interpretability. Unlike traditional dropout layers, which randomly set individual features to zero, the view dropout layer zeroes out all the input features of a single omic layer. This approach encouraged the model to reconstruct the data by learning the relationships among multiple omic layers, rather than relying on one specific omic layer. For example, in generating drug response predictions, the MOSA model could disproportionately emphasize the input drug response data, neglecting the potential contributions from other omic layers, such as transcriptomic and proteomic data. By using the view dropout layer, we significantly improved the latent space cell line separation (Fig. 1c, Supplementary Figs. 1b, 10a, b) and reconstruction for both the proteomic (Fig. 3a, Supplementary Fig. 10c) and drug response data (Supplementary Fig. 10d). The dropout rate for this layer is controlled by the hyperparameter view_dropout, which was optimally set as 0.5 for the final model.

**Model explanation via SHapley Additive exPlanations (SHAP)**. For model explanation, we used the Python package SHAP[22] (v0.42.1) with technical modifications to support the multi-omic data as the input to MOSA. Specifically, the GradientExplainer, which combines IntegratedGradient[68] and SmoothGrad[69], was used to calculate the changes of the gradients on the model's output regarding its input to attribute an importance value to each feature. The SHAP calculation was performed in two ways.

First, SHAP was run to explain the encoder part of MOSA, treating the integrated latent dimensions as the output. The result contains SHAP values in a multidimensional array with shapes of ($N_{latent\_dim}$, $N_{samples}$, $N_{features}$), where each $N$ represents the number of latent dimensions, samples and features, respectively. To achieve the global level feature importance for analysis, the multidimensional array was first taken as the absolute value to account for both positive and negative impact, and then summed across latent dimensions, followed by averaging by samples. This then resulted in a list of length $N_{features}$, representing the overall feature importance contributing to the latent space (Fig. 4a).

Second, SHAP was run to explain MOSA's reconstruction of each omic dataset. Taking drug response as an example, similarly to explaining the latent space, the shape of the SHAP values is ($N_{drugs}$, $N_{samples}$, $N_{features}$), where $N_{drugs}$ represents the number of drugs, and $N_{samples}$, $N_{features}$ are described as above. In this analysis, the array was only averaged across samples, resulting in a 2D array of ($N_{drugs}$, $N_{features}$), which measures the feature importance for each drug. The feature 1-methylnicotinamide metabolite was first selected and the drugs that had the highest SHAP values were analyzed to identify EMT-related drugs (Fig. 4b). Other important features for the drugs of interest were then ranked by selecting the row of the drug and then ranking the features in the descending order (Fig. 4c). Due to the limitation of the computational resource, 20% of the samples were randomly selected to compute the feature importance for reconstructing drug response and copy number datasets, while 20 samples were randomly selected for other omic datasets which have much larger number of dimensions.

Overall, the SHAP analysis allowed us to identify features that are important for the multi-omic latent dimension and for explaining the reconstruction of features, such as drug response. Feature importance aggregated across all the samples and output dimensions can be found in Supplementary Data 9 and 10. More granular feature importances for each output dimension can be downloaded from the figshare repository provided in the *code and data availability* section.

**Loss function**. The loss function is the summation of three components: 1) Reconstruction error across all input datasets; 2) weighted variational Kullback–Leibler (KL) regularization term of the multi-omic joint latent dimensions[70]; and 3) a contrastive loss using tissue types as labels:

$$Loss_{total} = L_{reconstruction} + \lambda L_{KL} + \alpha L_{contrastive} \quad (1)$$

The reconstruction loss $L_{reconstruction}$ is defined as:

$$L_{reconstruction} = \sum_d l_d \quad (2)$$

Where $l_d$ represents the reconstruction loss for dataset $d$, calculated using the mean squared error (MSE)[71,72].

In Eq. (1) the $\lambda$ and $\alpha$ are optimized hyperparameters to weight the KL divergence and contrastive loss terms, respectively. $L_{KL}$ calculates the KL divergence between the learned gaussian distribution with mean ($\mu$) and variance ($\sigma^2$) of the VAE and a standard normal prior distribution[70].

The last part of the loss function is a contrastive loss defined as:

$$L_{contrastive} = [m_{pos} - s_p]_+ + [s_n - m_{neg}]_+ \quad (3)$$

where $s_p$ and $s_n$ represents the cosine similarity between positive pairs and negative pairs, which are defined by whether two samples have the same tissue type. $m_{pos}$ and $m_{neg}$ are positive and negative margins, which are hyperparameters tuned as described in the section below.

**Asymmetrical VAE**. MOSA was also engineered with an asymmetrical structure to optimize model efficiency by reducing the number of parameters. Specifically, feature selection was conducted in a data-type-specific manner before the encoding process. For transcriptomic and methylation data, only features that exhibited high variability were selected as input to the model. Highly variable features were defined using a gaussian mixture model with two components fitted to the standard deviation of all features, thus capturing two distributions of lowly and highly variable features. The standard deviation threshold is defined as the biggest value at which the densities of the two distributions are equal, hence features with a standard deviation greater than 1.122 for transcriptomic data and 0.064 for methylation data are considered highly variable and selected as input for the encoder. For CRISPR-Cas9 data, gene knock-outs that did not significantly impact any cell line, as indicated by a gene fitness score higher than −0.5 in every cell line, were excluded from the input layer. This targeted feature selection effectively reduced the model's computational burden. Despite this reduction in input complexity, all available features were included during the decoding process to reconstruct the data. This asymmetrical design was chosen for its ability to maintain the model's predictive and reconstructive capacities while streamlining its architecture.

**Table 1 | Optimized hyperparameters used for training MOSA**

| Parameter | Value |
|---|---|
| Number of epochs | 500 |
| Batch size | 256 |
| Learning rate | 3e−4 |
| Number of cross-validation folds | 3 |
| Feature missing rate threshold (%) | 0.85 |
| Latent dimensions of each view (%) | 0.25 |
| Number of joint latent dimensions | 200 |
| Hidden dimensions (%) | 0.7 |
| Dropout probability | 0.4 |
| View dropout probability | 0.3 |
| Weight of Kullback–Leibler (KL) loss term | 0.0001 |
| Weight of the contrastive loss term | 0.005 |
| Positive margin of the contrastive loss | 0.85 |
| Negative margin of the contrastive loss | 0.15 |
| Optimizer | Adam |
| Weight decay | 5e−4 |
| Activation function | PReLU |
| Scheduler | Plateau |
| Scheduler threshold | 1e−4 |
| Scheduler factor | 0.6 |
| Scheduler patience | 7 |
| Scheduler minimum learning rate | 1e−7 |

**Hyperparameters.** The choice of hyperparameters (Table 1) was guided by an automatic optimization framework based on parallel trials (Optuna[73]) and then manually adjusted. For each run a stratified shuffle split is performed, stratifying by hematopoietic and lymphoid cell lines, leaving 20% of the samples for testing. A total of 600 trials were performed, where each trial was capped to 150 epochs.

### Benchmark state-of-the-art methods

For comparison with the unsupervised multi-omics approach taken by MOSA, 16 multi-omics integration methods were tested. However, as listed in Supplementary Data 6, for most of the models, we encountered issues related to intrinsic design and implementation choices. These included, for example, limitations on the number of supported omic modalities and specific designs tailored only for count data processing since many of the models are designed for single-cell data. Therefore, we have managed to run and systematically benchmark our results for all seven omics datasets considered against three other state-of-the-art methods for multi-omics data integration, including MOFA[10,14] as a linear multi-omics dimensionality reduction approach, MOVE[32] as a VAE-based approach, and mixOmics[35,36], which is based on generalized canonical correlation analysis. To make comparisons as close as possible and focus solely on methodological differences, the same data preprocessing was used for MOSA, MOFA, MOVE and mixOmics. Similarly, the number of factors for MOFA was initially set to the same optimal number of joint latent dimensions, i.e., 200 (Table 1). However, this generated poorly performant results, i.e., poorly reconstructed datasets. Through manual exploration, the optimal number of factors was set to 100, which was automatically reduced during training to 97 by discarding factors with variance explained lower than 0.0001. Each view was scaled independently, and the model was run until it converged (convergence_mode = slow). The number of dimensions was successfully set as 200 in MOVE. Since mixOmics requires the number of dimensions separately associated with each omic dataset, 210 dimensions were used in mixOmics to achieve the closest comparison. The conditionals layer in MOSA contains both binary, e.g., mutations and tissue of origin, and continuous,

e.g., cancer cell line growth rates and doubling times from independent studies, features. These data were included as a separate layer in MOVE, however, MOFA does not support a mixed distributed view, e.g., Gaussian and Bernoulli. Thus we could not integrate the growth rate and doubling time in the conditional view, which apart from these two have only binary features, and therefore the prior likelihood distribution was set to Bernoulli. These configurations produced the best multi-omic dimensionality reduction and view reconstruction using MOFA. The optimized model was saved as an HDF5 file and is also provided in the figshare repository. Similarly, the tissue-of-origin data was used as the target variable in mixOmics following the documentation of the package. Specifically, the DIABLO mode[36] (N-integration) in the mixOmics suite, which was an extension to the original mixOmics toolset[35], was used for the multi-omics data integration task in this study.

In order to evaluate MOSA more comprehensively and to address the limitations of many state-of-the-art methods that support only a limited number of omic modalities, we conducted a separate benchmarking analysis considering only two omic modalities as input. This approach allowed us to include four additional methods: JAMIE[18], scVAEIT[44], iClusterPlus[74] and moCluster[45]. Transcriptomic and drug response data were used to train and benchmark MOSA against seven other methods, as these omic types are the focus of our benchmarks. Since only two omics were included, we removed the requirement for a sample to have data from at least two omics. JAMIE, iClusterPlus and moCluster were successfully run using 200 dimensions for the integrated latent space, which was the same as MOSA. However, scVAEIT with 200 latent dimensions generated poor results, especially for the latent space clustering comparison. Similar to MOFA, we manually searched for the optimal setting and decided on using 100 latent dimensions for scVAEIT. Additionally, Gaussian distribution was set for the dist_block hyperparameter for both the transcriptomic and drug response data. MOSA, MOFA, MOVE, JAMIE and scVAEIT were evaluated for both synthetic data reconstruction and clustering performance, while mixOmics, iClusterPlus, and moCluster were only included in the latent space clustering comparison as they are not generative models. To ensure fair comparisons, conditionals were not incorporated into any of the selected methods during training.

### Protein-protein interaction co-abundance analysis

Protein-protein interactions (PPIs) were estimated using two methods to compare the ability of MOSA augmented proteomics matrix and the original proteomic matrix to recapitulate PPIs present in specific protein interaction resources datasets: CORUM[75], BioGRID[76] and STRING[77]. The first method, Pearson's r, has been previously used for this task[39]. Due to its inherent limitations, e.g., its inability to account for confounding effects and data structure, a new method, similar to that described by Wainberg et al.[40] based on a generalized linear model (GLM) was tested. This method applies Cholesky's Whitening transformation to proteomics data by using the inverse of its covariance matrix, which decorrelates samples and pushes data towards normality. This transformed data is then used in an ordinary least squares (OLS), whose calculated weights are a correlative metric between two proteins. Each method was calculated for every protein pair, with all pairs being ordered for each method by ascending p-value. Afterwards, a curve was drawn based on the cumulative sum of presence of that pair on a PPI set, either $\frac{1}{k}$ (presence) or 0 (absence), where $k$ is the total number of present pairs. Thus, the better the method, the greater the AUC of the recall curve.

### Statistics & reproducibility

Sample sizes were determined by the availability of cancer cell lines and the associated multi-omic datasets from the Cancer Dependency Map (DepMap). A total of 1523 cancer cell lines were included, for which at least two datasets were available. No statistical method was

used to predetermine sample size. Data exclusion details can be found under the Data preprocessing section in Methods. Data were randomly split for cross-validation purposes. The randomization was stratified by hematopoietic and lymphoid cell lines to ensure balanced representation across cell line types and distinct culture conditions, i.e. suspension vs adherent. Blinding was not applicable to this study, as all analyses were conducted using publicly available in vitro data from cancer cell lines. The findings were validated using independent datasets for proteomics (CCLE), drug response (GDSC and CTD2), and transcriptomics (DepMap). A 10-fold cross-validation strategy was applied to assess the reproducibility of the MOSA model across multiple omics layers.

## Inclusion and ethics

All authors have committed to upholding the principles of research ethics and inclusion as advocated by the Nature Portfolio journals.

## Reporting summary

Further information on research design is available in the Nature Portfolio Reporting Summary linked to this article.

## Data availability

All data were assembled from the Cancer DepMap and synthetic datasets generated have been deposited in figshare under the following URLS: DepMap datasets: https://doi.org/10.6084/m9.figshare.24420580. https://doi.org/10.6084/m9.figshare.24420598. MOSA augmented datasets and latent representation: https://doi.org/10.6084/m9.figshare.24562765. MOSA feature importance: https://doi.org/10.6084/m9.figshare.24473005. MOFA multi-omics reconstruction and latent representation: https://doi.org/10.6084/m9.figshare.24420631. MixOmics multi-omics latent representation: https://doi.org/10.6084/m9.figshare.25764408. MOVE diabetes multi-omics reconstruction and latent representation: https://doi.org/10.6084/m9.figshare.25764438.

## Code availability

All code is available at https://github.com/QuantitativeBiology/PhenPred (https://doi.org/10.5281/zenodo.13945138). The pretrained weights of MOSA are available at https://huggingface.co/QuantitativeBiology/MOSA_pretrained (https://doi.org/10.57967/hf/3634).

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

## Acknowledgements

We thank the Broad Institute and the Wellcome Sanger Institute for, through the Cancer Dependency Map consortium, making their data freely available and readily accessible to the scientific community and thereby enabling this work. This research was funded in part by the Wellcome Trust Grant 206194. ProCan® is supported by the Australian Cancer Research Foundation, Cancer Institute New South Wales (NSW) (2017/TPG001,REG171150), NSW Ministry of Health (CMP-01), The University of Sydney, Cancer Council NSW (IG 18-01), Ian Potter Foundation, the Medical Research Futures Fund (MRFF-PD), National Health and Medical Research Council (NHMRC) of Australia European Union grant (GNT1170739, a companion grant to support the European Commission's Horizon 2020 Program, H2020-SC1-DTH-2018-1,'iPC- individualized Paediatric Cure' [ref. 826121]), and National Breast Cancer Foundation (IIRS-18-164). Work at ProCan® is done under the auspices of a Memorandum of Understanding between Children's Medical Research Institute and the U.S. National Cancer Institute's International Cancer Proteogenome Consortium (ICPC), that encourages cooperation among institutions and nations in proteogenomic cancer research in which datasets are made available to the public. Z.C. is the recipient of a PhD Scholarship from Sydney Cancer Partners with funding from Cancer Institute NSW (2021/CBG0002). A.R.B. is funded by the Portuguese national agency Fundação para a Ciência e a Tecnologia (FCT) through the research grant UI/BD/154599/2022. This work has received funding from the European Union's Horizon 2020 research and innovation program under grant agreement no. 951970 (OLISSIPO project). For open access, the authors have applied a CC BY public copyright license to any Author Accepted Manuscript version arising from this submission. This work was supported by national funds through FCT, under project UIDB/50021/2020 (https://doi.org/10.54499/UIDB/50021/2020). The authors acknowledge the OSCARS project, funded by the European Commission's Horizon Europe Research and Innovation Programme under grant agreement No. 101129751.

## Author contributions

Z.C., S.A., A.R.B., M.D.S., C.P. and E.G. implemented analyses. Z.C., S.A. and E.G. wrote the software. E.G. supervised and conceptualized the study. S.V., P.J.R., R.R.R., M.J.G, Q.Z. and E.G. acquired funding and contributed to methodology. Z.C, S.A., A.R.B., Q.Z. and E.G. wrote the manuscript. All authors have revised and approved the manuscript.

## Competing interests

AstraZeneca, GlaxoSmithKline, and Astex Pharmaceuticals have awarded M.J.G. research grants and M.J.G. is founder and advisor at Mosaic Therapeutics. All other authors declare no competing interests.
