## [Transparent Peer Review file · Nature Communications]

Synthetic augmentation of cancer cell line multi-omic datasets using unsupervised deep learning

Corresponding Author: Professor Emanuel Goncalves

Version 0:

Reviewer comments:

Reviewer #1

(Remarks to the Author)

This is an interesting study in the area of data augmentation due to data complexity and sparsity. The authors propose an unsupervised deep learning model, MOVE (Multi-Omic Variational Encoder), to integrate and augment DepMap. While the synthetic data generate a significant number of multi-omic profiles, this study does have some limitations as discussed below.

Major points:

1. By employing the data augmentation technique, one can artificially increase the size of the dataset (i.e., derive new artificial data) from the existing dataset. This technique employs various operations such as sequence replacement, sequence flipping, and sequence cropping to improve the prediction performance. The data augmentation technique aids in overcoming the overfitting and data limitation issues. Thus, employing this technique will ensure precision, accuracy, and reliability eventually leading to better performance. There are some literature papers in this field already (e.g., <https://www.frontiersin.org/journals/genetics/articles/10.3389/fgene.2023.1199087/full>) The authors should include all recent work in their introduction.

2. Several recent studies reported new methods on multi-modal data augmentation, such as:

<https://arxiv.org/abs/2212.14453>

<https://arxiv.org/abs/2206.08358>

How does this method compare with these published tools?

3. For data augmentation, there is a recent work on DeepBIO (Ruheng Wang et al, Nucleic Acids Research, Volume 51, Issue 7, 24 April 2023).

They have also designed a section for data augmentation to pre-process the input data. Can the authors compare to this method?

4. Multi-omics data is being generated in many areas of biological functions, including transcriptomics data and drug response data, such as Kircher et al 2022 (<https://www.mdpi.com/1422-0067/23/5/2481>) and Partin et al 2023 (<https://www.frontiersin.org/articles/10.3389/fmed.2023.1058919/full>). It would be important to compare with these methods, and show how the advantage of the current method.

5. "Learning-augmented assessment" workflow (PMID: 30467458) has been proposed on multi-omics data to predict drug response, etc. The comparison with these previous tools are critical to demonstrate the novelty/advances of this method.

6. The authors show an increase of 32.7% in the number of multi-omic profiles and generated a DepMap for 1,523 cancer cell lines. It would be helpful to use an independent dataset (such as PPI network data where it's widely known to have a large false negative rate) to show to applicability across datasets.

7. The authors claim that their synthetically enhanced data uncovers less studied mechanisms associated with drug resistance, and biomarker identification related to drug and gene dependencies. It would be important to validate some of the predicted synthetic data, by wet-lab experiments, even in a number of cell lines. That would help validate the model.

Minor points:

1. Many commas are used in the middle of sentences, which decreases clarity, such as "This raises the importance of

developing holistic machine learning models capable of integrating, beyond genomics, other types of omics” on Page 2. The authors should correct such cases.

2. Some grammatic and writing style issues:

For example, this sentence is too long for readers: “We systematically evaluated and benchmarked MOVE, demonstrating its 62 capabilities in reconstructing independent drug response and proteomic datasets, recovering cancer cell line tissue and cell type clustering, and increasing statistical power to find genomic associations with CRISPR-Cas9 gene essentiality screens.”

It would be the best to break this into more sentences, or consider using clauses.

Reviewer #2

(Remarks to the Author)

Summary:

This study presents an advancement in the field of multi-omics data integration, particularly for cancer cell lines, by employing a Variational Autoencoder (VAE) approach with an innovative asymmetrical design. Notably, the research extends the integration to seven different modalities, surpassing the typical range of 2-3 modalities in existing studies. This broader scope could yield substantial insights into cancer genomics.

Noteworthy Results:

The most significant achievement of this study is the comprehensive coverage of seven omics modalities. This extensive integration marks a notable progression from the norm and could provide a more holistic understanding of cancer genomics. The asymmetrical design of the VAE is another strength, enhancing the efficiency of optimization. The downstream analysis fortifies the credibility of the imputation results.

Significance to the Field:

The study is poised to make a meaningful contribution, offering imputed data that can be directly applied in downstream tasks. However, the methodology, primarily based on VAE, aligns closely with established practices in the field. While the application of VAE in imputing omics data isn't novel, the comprehensive integration of multiple modalities enhances the value of this work.

Evidence Supporting Conclusions:

The paper provides ample evidence to substantiate its claims, demonstrating a robust approach to data imputation in multi-omics studies.

Data Analysis, Interpretation, and Conclusion:

The primary concern lies in the evaluation approach. The study employs relatively basic methods for comparison, such as mean imputation or linear models. Given the maturity of VAE applications in omics data imputation, a more rigorous comparison with state-of-the-art methodologies is warranted. For instance, comprehensive surveys like the ones found in NCBI and Genome Biology detail over ten different papers each, offering a variety of approaches in this domain.

Methodological Soundness:

The study assumes data missing at random, which might not always be the case in omics studies, as some missing data could result from systematic biases. This assumption potentially limits the applicability of the learned VAE model, as it might not accurately impute data missing due to non-random factors.

Reproducibility:

The study provides sufficient detail in its methods, facilitating reproducibility and further application in the field. The study also provided the imputed data.

Version 1:

Reviewer comments:

Reviewer #2

(Remarks to the Author)

The revision addresses the concerns on the previous work that serves as a baseline. The comprehensiveness is impressive. The supporting experimental results on testing the missing not at random is also reasonable and support its claim.

Reviewer #3

(Remarks to the Author)

The article by Zhaoxiang Cai et al. introduces a pipeline named MOSA for integration and imputation of Multi-omics data. MOSA was then used on DepMap data for augmentation of the molecular and phenotypic profiles. The authors then propose that MOSA can be used for biomarker and drug target identifications.

I agree with the previous reviewer that this study lacks systematic benchmarking by comparing with the existing tools. Experimental validations, which are completely missing, are also essential for the authors' claims. Therefore, I am afraid that I don't think the reviewers' concerns have been satisfactorily addressed. I am not convinced by the authors' claims that most of the existing tools are not feasible for comparison and that experimental validations are not needed.

If the paper is positioned as a method paper, MOSA has to be extensively tested and compared with ALL the existing tools currently in use. Of course, most of the existing methods were not designed to handle all 7 types of omics data, but that does not necessarily mean that they are not worth comparing with. Simply put, if MOSA is only usable for DepMap data, this study would not be suitable for publication as a method paper due to the limited scope.

If the paper is positioned as an analysis type of article, the authors would need to present the new insights and novel findings from their study. In addition, the analysis results should be presented in a more user-friendly way, for example, a web server that allows user to freely explore the imputation and original data and mine for biological insights based on their interests, like what DepMap has done.

Nevertheless, certain levels of experimental validations of the key discoveries from the pipeline are surely needed to prove that the analysis pipeline of MOSA makes sense in biology and is of unique value for the community.

Version 2:

Reviewer comments:

Reviewer #3

(Remarks to the Author)

1. I think that the authors should include a comprehensive evaluation of the existing methods in the main text. In other words, information in supplementary table 10 should be presented in the main text.

2. I am still not convinced that MOSA can only be compared with other methods that can handle all the 7 omics data. It should be done, and can be done, to compare the data augmentation results of MOSA (with multi-omics data) with the results of other methods (for example JAMIE, which only used 2 omics). Again, the authors need to explicitly show that MOSA outperforms the others even if they only handles smaller datasets.

Version 3:

Reviewer comments:

Reviewer #3

(Remarks to the Author)

The authors have added more benchmarking of their pipeline by comparing MOSA with other existing tools. I appreciate their efforts, and I have no further concerns.

Point-by-point response to the reviewers' comments

We thank the reviewers for their helpful comments and have addressed their suggestions (*in italic*) in a point-by-point response written in blue.

Please note that we chose to change the name of our method to avoid conflict with an existing variational autoencoder for multi-omics integration, known as MOVE, to which we now also benchmark. We have renamed our approach as **MOSA** (Multi-Omic Synthetic Augmentation), which also emphasizes the synthetic data augmentation aspect of our approach.

Reviewer #1 (Remarks to the Author): expertise in multi-omics data integration

This is an interesting study in the area of data augmentation due to data complexity and sparsity. The authors propose an unsupervised deep learning model, MOVE (Multi-Omic Variational Encoder), to integrate and augment DepMap. While the synthetic data generate a significant number of multi-omic profiles, this study does have some limitations as discussed below.

Major points:

1. By employing the data augmentation technique, one can artificially increase the size of the dataset (i.e., derive new artificial data) from the existing dataset. This technique employs various operations such as sequence replacement, sequence flipping, and sequence cropping to improve the prediction performance. The data augmentation technique aids in overcoming the overfitting and data limitation issues. Thus, employing this technique will ensure precision, accuracy, and reliability eventually leading to better performance. There are some literature papers in this field already (e.g., <https://www.frontiersin.org/journals/genetics/articles/10.3389/fgene.2023.1199087/full>) The authors should include all recent work in their introduction.

We thank the reviewer for the reference and we have expanded the introduction to cover the different aspects mentioned.

“Despite recent successes of deep learning¹ multi-omic integration faces several limitations, most importantly, high heterogeneity of different data types (e.g., discrete vs. continuous distributions), intrinsic technological limitations (e.g., missing values), and limited data availability (e.g., in this study, only 25.8% of the cancer cell lines have a complete set of all seven omic datasets under consideration)².”

“Moreover, VAEs have highly flexible designs that can handle data sparsity robustly, and are easily extensible to incorporate different data types. In particular, methods based on VAE models have demonstrated significant success in the field of single-cell multi-omics integration and augmentation. However these methods often presuppose the presence of specific data types, such as count data from scRNA-seq and scATAC-seq, limiting their applicability across broader omic landscapes³⁻⁷ .”

2. Several recent studies reported new methods on multi-modal data augmentation, such as:

<https://arxiv.org/abs/2212.14453>

<https://arxiv.org/abs/2206.08358>

How does this method compare with these published tools?

We thank the reviewer for the two references related to multi-modal data augmentation. The MixGen⁸ enhanced training for vision-language models by merging images and texts. LeMDA⁹ demonstrated superior performance relative to MixGen and was utilized across various prediction tasks. The multi-modal data in these two papers refer to multiple modalities such as image, text and tabular data, with both methods concentrating on training strategies rather than on a specific neural network architecture. In the meantime, our model prioritizes tabular data from multiple views (omics), and is tailored to the Cancer DepMap with modifications to the variational autoencoder architecture. Several other approaches^{5-7,10-13}, besides those mentioned by the reviewer, have been recently proposed, albeit two key limitations are common: 1) these are mostly developed for single-cell studies and 2) do not integrate more than three omics (see more details in **response 1.4**).

We now discuss these two papers in the future work section of the manuscript:

“In addition to tabular omic data, VAEs have demonstrated great successes integrating image and text-based data^{8,9}, and MOSA can be further enhanced to integrate these type data and enable multi-modal data augmentation.”

3. For data augmentation, there is a recent work on DeepBIO (Ruheng Wang et al, *Nucleic Acids Research*, Volume 51, Issue 7, 24 April 2023). They have also designed a section for data augmentation to pre-process the input data. Can the authors compare to this method?

The data augmentation method used in DeepBIO works with sequence data specifically, while MOSA uses the processed tabular data as the input. The techniques used in DeepBIO include sequence replacement, sequence flipping and sequence cropping, which are similar to the pre-processing techniques for image classification, such as random cropping, rotation and flipping. MOSA is particularly designed for the Cancer DepMap tabular data.

4. Multi-omics data is being generated in many areas of biological functions, including transcriptomics data and drug response data, such as Kircher et al 2022 (<https://www.mdpi.com/1422-0067/23/5/2481>) and Partin et al 2023 (<https://www.frontiersin.org/articles/10.3389/fmed.2023.1058919/full>). It would be important to compare with these methods, and show how the advantage of the current method.

The primary advantage of MOSA, in comparison to the two studies mentioned, as well as other existing methods, lies in its capability to vertically integrate data from seven distinct omics and thereby leverage relationships between these omics for synthetic data generation. MOSA can produce data for cell lines that entirely lack specific omics by leveraging the other layers.

Both studies mentioned focus on data augmentation within a single type of omic data. Kircher et al.¹⁴ evaluated three methods for generating synthetic transcriptomic data solely from gene expression data as the input, thus missing potential associations across omics. For example, by integrating genomics and CRISPR-Cas9 gene essentiality,

we can successfully capture the association between FLI1-EWSR1 fusion and the dependency on FLI1, showing that synthetically generated gene essentiality screens of cell lines without CRISPR-Cas9 actual measured data, recapitulate the expected strong dependency on cells that harbor the FLI1-EWSR1 fusion (**Figure 3f**). This is particularly notable considering we only have n=23 FLI1-EWSR1 fusions to train the model. Partin et al.¹⁵, similar to DeepBIO, applied data augmentation techniques tailored to the specific format of data used in their study. For instance, the authors incorporated the chemical structure of drugs as part of the features for predicting responses. In contrast, MOSA does not integrate compound structure information but rather the drug response measurements (i.e. IC50s, the concentration of the drug necessary to reduce in 50% the viability of the cancer cell lines) as features in an unsupervised manner.

We agree with the reviewer that it is important to more broadly benchmark our approach with other existing methods. Based on this suggestion, we have expanded our analyses to now include a benchmark with two additional methods, MOVE¹⁶ and mixOmics (using the latest DIABLO mode)^{17,18}, which are also developed to integrate multiple omics. While at the end we only added two methods, we searched and compiled a comprehensive list of 14 methods, but many had intrinsic limitations that hampered the application to our Cancer DepMap datasets (**Revision Table 1**).

Methods	Notes
General multi-omics methods	
OmicAnalyst ¹⁹	It is a web-based tool and only supports 5 omic datasets.
MOVE¹⁶	We managed to adapt and add it to the comparisons with a reasonable amount of code changes to export results.
OmiEmbed ²⁰	The tool has datasets hard coded, limiting the generalisability to more omic datasets. Furthermore, the method is only unsupervised for its Phase 1, and the code requires data for a downstream (regression,

classification or survival) to run. Since our analysis is fully unsupervised, we were not able to run this method.

https://github.com/zhangxiaoyu11/OmiEmbed/blob/main/datasets/abc_dataset.py

mixOmics/DIABLO^{17,18}

We have incorporated mixOmics into our comparative analysis. While mixOmics is not among the most recent tools available, we chose to include it because it is widely used in the field.

iClusterBayes²¹

It only supports 6 omic datasets. Source code not available.

iClusterPlus²²

It only supports 4 omic datasets. Source code not available.

Single-cell multi-omics methods

MIDAS⁵

MIDAS is explicitly designed to process count data, with a targeted application for RNA-seq, ADT (Antibody-Derived Tags), and ATAC-seq datasets. This requirement cannot be met by our study because of, for example, the drug response data. Despite our attempts to adapt MIDAS to fit the broad and varied requirements of the DepMap datasets, we were unable to run it.

StabMap⁶

StabMap was not originally conceived to accommodate the integration of data on the scale presented by the DepMap datasets. In an effort to align with the methodology prescribed by the authors, we partitioned the samples into 52 distinct sets, each representing a different combination of omic data types. The scale diverges significantly from that of the original study, which only considered three sets derived from two types of omic data. Subsequently, we encountered a critical error stating "feature network is not connected,

features must overlap in some way via rownames".

JAMIE⁴

We managed to run the tool, but the method was developed to run for datasets with two omics.

<https://github.com/daifengwanglab/JAMIE/blob/main/jamie/jamie.py#L408>

MultiVI and other scVI tools⁷

The suite of scVI tools is specifically designed to process raw count data, as the generative component of the model employs either the negative binomial or Poisson distribution to accommodate the inherent properties of such data (<https://docs.scvi-tools.org/en/stable/api/reference/scvi.model.MULTIVI.html>). Moreover, MultiVI, the tool within the scVI series that is designed for multi-omics integration, explicitly necessitates ATAC-seq data as a prerequisite for analysis. Given that ATAC-seq data is absent from the DepMap project's dataset, this condition further limits the applicability of MultiVI for our purposes.

scVAEIT¹²

scVAEIT is designed to jointly analyze multi-modal single-cell datasets (CITE-seq, ASAP-seq, and DOGMA-seq), performing mosaic integration. However, with the DepMap datasets we are performing vertical integration, considering only different views. Therefore, despite the adaptability of this model to other datasets and probability distributions, an error was encountered regarding the subconnection level block, including the `dim_block` parameter. This precluded the integration of scVAEIT in our benchmarks. However, this model should be considered as a state-of-the-art tool for mosaic integration tasks due to its flexibility in adapting the code to other single-cell datasets.

Multigrade¹¹

Multigrade is also explicitly designed to process count

	data, as evidenced by its core implementation which fundamentally relies on count-based statistical distributions for its analyses (source code: https://github.com/theislab/multigrade/blob/649bdfcfc587bd4abf7f7b8a1dea98d5101d653/src/multigrade/model/_multivae.py#L370).
scMM³	The scMM framework, as currently implemented, offers VAE sub-classes exclusively tailored to ATAC, Protein, and RNA data types (https://github.com/kodaim1115/scMM/tree/master/src/models). Furthermore, the associated publication primarily investigates dual-omics scenarios, specifically CITE-seq and SHARE-seq. This focus on dual-omics, alongside the restriction to only three types of omic data, presents a significant limitation for our study, which necessitates a broader and more versatile analytical approach to accommodate the seven different omic datasets within the DepMap project.
Cobolt²³	The Cobolt framework includes a latent model for each type of omics (in the paper, only single-cell RNA-seq and ATAC-seq datasets are considered) inspired by the Latent Dirichlet Allocation. It assumes a generative distribution for the input data where the counts measured on a cell are the mixture of the counts from different latent categories. Therefore, this model is designed specifically for count data (source code: https://github.com/epurdom/cobolt/blob/70b6ff1365c4fb2161e4297f8455455e9e87354/cobolt/utils/data.py#L2).

Revision Table 1. Summary of Considered Methods.

Having managed to adapt and run MOVE and mixOmics to integrate our datasets, we have assessed and compared their performance, alongside MOFA, including clustering in the multi-omic latent space (**Figure 1c, Supplementary Figure 1**), and synthetic data

generation for unseen drug response data (**Figure 2b-e**). In comparison to MOFA, MOVE, and mixOmics, our MOSA model demonstrated improved separation by tissue types in UMAP visualizations of the latent joint space (**Figure 1c, Supplementary Figure 1a-c**). This enhanced performance was further quantified using the Calinski-Harabasz and the Davies-Bouldin scores for clustering evaluation (**Supplementary Figure 1d**). Next, we focused on the performance of these tools in reconstructing a newly generated, independent drug response dataset (**Figure 2b-e**). While mixOmics allows for the integration of multi-omic data with missing values, it does not support whole-omic. Therefore, we only included MOVE and MOFA+ in this particular comparison (**Figure 2d**). Notably, the data generated by MOVE exhibited limited variability from zero, which resulted in suboptimal performance across all three evaluation metrics used: Pearson's r , Spearman's ρ , and mean squared error (MSE). This issue may stem from the architectural limitations of MOVE, which lack omic-specific encoders and decoders.

Figure 1. Cancer multi-omic integration with MOSA. a) cancer cell line multi-omic datasets integrated. Purple represents measured datasets, while orange represents gaps, i.e., missing datasets, across the 1,523 cancer cell lines. b) schematic of the autoencoder, MOSA, where encoders are represented at the top and decoders at the bottom. For simplicity, the integration of only two datasets is represented. Highlighted designs of MOSA are illustrated on the right. c) dimensionality reduction visualized using Uniform Manifold Approximation and Projection (UMAP) representation of the trained MOSA joint latent space, where each dot represents a cancer cell line colored according to its tissue of origin.

Supplementary Figure 1. Latent space visualization comparison. a) UMAP representation of the trained MOFA joint latent space, where each dot represents a cancer cell line and is coloured according to its tissue of origin. b) and c), UMAP representations for MOVE and mixOmics, respectively. d) comparison of cell line separations quantified by Calinski-Harabasz index (higher value indicates better) and Davies-Bouldin index (lower value indicates better).

Figure 2. Reconstruction of MOSA for drug response and CRISPR-Cas9 datasets. **a)** MOSA reconstruction quality measured using a 10-fold cross-validation. After reconstructing all test folds, they are concatenated and the reconstruction quality score is calculated as the Pearson's r between the reconstructed and measured values. Features ranked by their reconstruction quality are shown for the drug response (left) and the CRISPR-Cas9 (right) datasets. Duplicated drug names represent replicated screens for the same drug. Representative examples of strongly selective CRISPR-Cas9 and drug responses are labeled. **b)** MOSA's partial dataset augmentation (missing value imputation) of drug IC50s compared to novel drug response screens. **c-e)**, similar to **b)**, using MOVE, MOFA and mean imputed values, respectively.

5. *“Learning-augmented assessment” workflow (PMID: 30467458) has been proposed on multi-omics data to predict drug response, etc. The comparison with these previous tools are critical to demonstrate the novelty/advances of this method.*

The study by Athreya et al.²⁴ predicts drug response in major depressive disorder utilizing a supervised model, but the method is not readily suited for data augmentation as it is designed as a supervised model. With respect to multi-omic data integration, this method used genome-wide association studies (GWAS) for feature selection, with the selected features being directly concatenated to form the integrated dataset. This contrasts with MOSA that seeks to derive a joint representation in the latent space from diverse omics, which is subsequently utilized by the decoders to generate synthetic data. We believe that due to their distinct applications, these two methods are not directly comparable in terms of their performance. MOSA's approach to learning a unified representation facilitates a more sophisticated integration of multi-omic data, potentially leading to more insightful synthetic data generation, distinct from the concatenation method used by²⁴.

6. *The authors show an increase of 32.7% in the number of multi-omic profiles and generated a DepMap for 1,523 cancer cell lines. It would be helpful to use an independent dataset (such as PPI network data where it's widely known to have a large false negative rate) to show to applicability across datasets.*

Indeed there are large databases of PPIs across many organisms and these have been tremendously informative. We have explored the potential of recapitulating PPI using the augmented dataset vs the original across three different databases, CORUM²⁵, STRING²⁶ and BioGrid²⁷). We observed that the augmented dataset shows a recall of PPI very similar to the original, except for CORUM where the original data outperforms. Albeit, since the original dataset contains missing values it precludes its application to certain types of models, i.e. generalized linear models (GLMs) across all ~1,500 cancer cell lines. We showed that with GLMs, the augmented dataset has performed similarly or outperformed the original dataset, including in the CORUM PPIs (**Supplementary Figure 5d**).

7. *The authors claim that their synthetically enhanced data uncovers less studied mechanisms associated with drug resistance, and biomarker identification related to drug*

and gene dependencies. It would be important to validate some of the predicted synthetic data, by wet-lab experiments, even in a number of cell lines. That would help validate the model.

We completely agree with the reviewer. Experimental validations are essential to confirm our hypotheses and if they validate across different cancer cellular models. The primary constraints are related to the extensive time and substantial financial resources required to conduct these experiments, which we believe we cannot consider within the scope of this manuscript. The diverse large-scale multi-omic and cross-institution datasets used to train and validate our model and hypotheses provide strong evidence to the associations identified, and hopefully put forward those that are most likely to validate experimentally. To clarify this point we added the following text to our discussion:

“Furthermore, while this has provided strong initial support for the EMT-related associations, further experimental work is necessary to validate and confirm these findings across different cancer cell models.”

Minor points:

1. Many commas are used in the middle of sentences, which decreases clarity, such as “This raises the importance of developing holistic machine learning models capable of integrating, beyond genomics, other types of omics” on Page 2. The authors should correct such cases.

We have edited the sentence as follows:

“This highlights the importance of developing holistic machine learning models capable of vertically integrating orthogonal datasets.”

We have also reviewed the manuscript for similar issues and made adjustments to ensure clarity throughout the text where necessary.

2. Some grammatic and writing style issues:

For example, this sentence is too long for readers: “We systematically evaluated and benchmarked MOVE, demonstrating its 62 capabilities in reconstructing independent drug response and proteomic datasets, recovering cancer cell line tissue and cell type clustering, and increasing statistical power to find genomic associations with CRISPR-Cas9 gene essentiality screens.”

It would be the best to break this into more sentences, or consider using clauses.

We have broken down the sentence as:

“In our study, we systematically evaluated and benchmarked MOSA, demonstrating its generative capacity across independent drug response and proteomic datasets and accurately recovering cancer tissue-of-origin clustering. Additionally, MOSA increased the statistical power to find genomic associations with CRISPR-Cas9 gene essentiality screens. Synthetically screened cancer cell lines revealed vulnerabilities consistent with genomic profiles, such as FLI1-EWSR1 fusion dependency.”

We have also carefully reviewed the rest of the manuscript and updated the text where necessary. This ensures that our findings are communicated clearly and concisely, enhancing the overall readability of our work.

Reviewer #2 (Remarks to the Author): expertise in multi-omics data integration

Summary:

This study presents an advancement in the field of multi-omics data integration, particularly for cancer cell lines, by employing a Variational Autoencoder (VAE) approach with an innovative asymmetrical design. Notably, the research extends the integration to seven different modalities, surpassing the typical range of 2-3 modalities in existing studies. This broader scope could yield substantial insights into cancer genomics.

Noteworthy Results:

The most significant achievement of this study is the comprehensive coverage of seven omics modalities. This extensive integration marks a notable progression from the norm and could provide a more holistic understanding of cancer genomics. The asymmetrical design of the VAE is another strength, enhancing the efficiency of optimization. The downstream analysis fortifies the credibility of the imputation results.

Significance to the Field:

The study is poised to make a meaningful contribution, offering imputed data that can be directly applied in downstream tasks. However, the methodology, primarily based on VAE, aligns closely with established practices in the field. While the application of VAE in imputing omics data isn't novel, the comprehensive integration of multiple modalities enhances the value of this work.

Evidence Supporting Conclusions:

The paper provides ample evidence to substantiate its claims, demonstrating a robust approach to data imputation in multi-omics studies.

We thank the reviewer for the positive comments, indeed the vertical integration of all broadly available omics in the Cancer DepMap was our major focus and novelty.

Data Analysis, Interpretation, and Conclusion:

The primary concern lies in the evaluation approach. The study employs relatively basic methods for comparison, such as mean imputation or linear models. Given the maturity of VAE applications in omics data imputation, a more rigorous comparison with state-of-the-art methodologies is warranted. For instance, comprehensive surveys like the ones found in NCBI and Genome Biology detail over ten different papers each, offering a variety of approaches in this domain.

We recognise the critical importance of rigorously benchmarking our Multi-Omics Synthetic Augmentation (MOSA) model against state-of-the-art methods. In the initial submission, our decision to include only MOFA+¹⁰ was driven by the fact that many recent VAE-based models are specifically developed for single-cell data, and thereby often cannot be directly used with bulk multi-omics datasets. For example, the DepMap datasets used in our study includes seven different omics, covering both continuous, binary and count data.

State-of-the-art single-cell multi-omic integration methods often assume simply the presence of specific types of omic data, such as scATAC-seq and scRNA-seq, and are not designed to integrate this diverse set of datasets.

In response to your feedback, we have dedicated a considerable effort to adapting existing tools for our large-scale datasets. Upon reviewing 14 methods (Revision Table 1), only MOVE¹⁶ and mixOmics (using the latest DIABLO mode)^{17,18} proved viable for the scale and diversity of our datasets, in addition to MOFA+¹⁰. We have documented the feasibility and limitations of each investigated method, providing clear reasons for their suitability or incompatibility with our data (Revision Table 1).

Methods	Notes
General multi-omics methods	
OmicAnalyst ¹⁹	It is a web-based tool and only supports 5 omic datasets.
MOVE¹⁶	We managed to adapt and add it to the comparisons with a reasonable amount of code changes to export results.
OmiEmbed ²⁰	The tool has datasets hard coded, limiting the generalisability to more omic datasets. Furthermore, the method is only unsupervised for its Phase 1, and the code requires data for a downstream (regression, classification or survival) to run. Since our analysis is fully unsupervised, we were not able to run this method. https://github.com/zhangxiaoyu11/OmiEmbed/blob/main/datasets/abc_dataset.py
mixOmics/DIABLO^{17,18}	We have incorporated mixOmics into our comparative analysis. While mixOmics is not among the most recent tools available, we chose to include it because it is widely used in the field.

iClusterBayes ²¹	It only supports 6 omic datasets. Source code not available.
--

iClusterPlus ²²	It only supports 4 omic datasets. Source code not available.
--

Single-cell multi-omics methods

MIDAS ⁵	MIDAS is explicitly designed to process count data, with a targeted application for RNA-seq, ADT (Antibody-Derived Tags), and ATAC-seq datasets. This requirement cannot be met by our study because of, for example, the drug response data. Despite our attempts to adapt MIDAS to fit the broad and varied requirements of the Cancer DepMap dataset, we were unable to make it work.
--

StabMap ⁶	StabMap was not originally conceived to accommodate the integration of data on the scale presented by the DepMap datasets. In an effort to align with the methodology prescribed by the authors, we partitioned the samples into 52 distinct sets, each representing a different combination of omic data types. The scale diverges significantly from that of the original study, which only considered three sets derived from two types of omic data. Subsequently, we encountered a critical error stating "feature network is not connected, features must overlap in some way via rownames".
--

JAMIE ⁴	We managed to run the tool, but the method was developed to run for datasets with two omics.
--

<https://github.com/daifengwanglab/JAMIE/blob/main/jamie/jamie.py#L408>

MultiVI and other scVI tools ⁷	The suite of scVI tools is specifically designed to process raw count data, as the generative component of the model employs either the negative binomial or
---	--

Poisson distribution to accommodate the inherent properties of such data (<https://docs.scvi-tools.org/en/stable/api/reference/scvi.model.MULTIVI.html>). Moreover, MultiVI, the tool within the scVI series that is designed for multi-omics integration, explicitly necessitates ATAC-seq data as a prerequisite for analysis. Given that ATAC-seq data is absent from the DepMap project's dataset, this condition further limits the applicability of MultiVI for our purposes.

scVAEIT¹²

scVAEIT is designed to jointly analyze multi-modal single-cell datasets (CITE-seq, ASAP-seq, and DOGMA-seq), performing mosaic integration. However, with the DepMap datasets we are performing vertical integration, considering only different views. Therefore, despite the adaptability of this model to other datasets and probability distributions, an error was encountered regarding the subconnection level block, including the `dim_block` parameter. This precluded the integration of scVAEIT in our benchmarks. However, this model should be considered as a state-of-the-art tool for mosaic integration tasks due to its flexibility in adapting the code to other single-cell datasets.

Multigrade¹¹

Multigrade is also explicitly designed to process count data, as evidenced by its core implementation which fundamentally relies on count-based statistical distributions for its analyses (source code: https://github.com/theislab/multigrade/blob/649bdfcfc587bd4abf7f7b8a1dea98d5101d653/src/multigrade/model/_multivae.py#L370).

scMM³

The scMM framework, as currently implemented, offers VAE sub-classes exclusively tailored to ATAC, Protein, and RNA data types (<https://github.com/kodaim115/scMM/tree/master/src/>

models). Furthermore, the associated publication primarily investigates dual-omics scenarios, specifically CITE-seq and SHARE-seq. This focus on dual-omics, alongside the restriction to only three types of omic data, presents a significant limitation for our study, which necessitates a broader and more versatile analytical approach to accommodate the seven different omic datasets within the DepMap project.

Cobolt²³

The Cobolt framework includes a latent model for each type of omics (in the paper, only single-cell RNA-seq and ATAC-seq datasets are considered) inspired by the Latent Dirichlet Allocation. It assumes a generative distribution for the input data where the counts measured on a cell are the mixture of the counts from different latent categories. Therefore, this model is designed specifically for count data (source code: <https://github.com/epurdom/cobolt/blob/70b6ff1365c4fb-d2161e4297f8455455e9e87354/cobolt/utils/data.py#L2>).

Revision Table 1. Summary of Considered Methods

Having managed to adapt and run MOVE and mixOmics to integrate our datasets, we have assessed and compared their performance, alongside MOFA, including clustering in the multi-omic latent space (**Figure 1c, Supplementary Figure 1**), and synthetic data generation for unseen drug response data (**Figure 2b-e**). In comparison to MOFA, MOVE, and mixOmics, our MOSA model demonstrated improved separation by tissue types in UMAP visualizations of the latent joint space (**Figure 1c, Supplementary Figure 1a-c**). This enhanced performance was further quantified using the Calinski-Harabasz and the Davies-Bouldin scores for clustering evaluation (**Supplementary Figure 1d**). Next, we focused on the performance of these tools in reconstructing a newly generated, independent drug response dataset (**Figure 2b-e**). While mixOmics allows for the integration of multi-omic data with missing values, it does not support whole-omic. Therefore, we only included MOVE and MOFA+ in this particular comparison (**Figure 2d**). Notably, the data generated by MOVE exhibited limited variability from zero, which resulted in suboptimal

performance across all three evaluation metrics used: Pearson's r , Spearman's ρ , and mean squared error (MSE). This issue may stem from the architectural limitations of MOVE, which lack omic-specific encoders and decoders.

Figure 1. Cancer multi-omic integration with MOSA. a) cancer cell line multi-omic datasets integrated. Purple represents measured datasets, while orange represents gaps, i.e., missing datasets, across the 1,523 cancer cell lines. b) schematic of the autoencoder, MOSA, where encoders are represented at the top and decoders at the bottom. For simplicity, the integration of only two datasets is represented. Highlighted designs of MOSA are illustrated on the right. c) dimensionality reduction visualized using Uniform Manifold Approximation and Projection (UMAP) representation of the trained MOSA joint latent space, where each dot represents a cancer cell line colored according to its tissue of origin.

Supplementary Figure 1. Latent space visualization comparison. a) UMAP representation of the trained MOFA joint latent space, where each dot represents a cancer cell line and is coloured according to its tissue of origin. b) and c), UMAP representations for MOVE and mixOmics, respectively. d) comparison of cell line separations quantified by Calinski-Harabasz index (higher value indicates better) and Davies-Bouldin index (lower value indicates better).

Figure 2. Reconstruction of MOSA for drug response and CRISPR-Cas9 datasets. **a)** MOSA reconstruction quality measured using a 10-fold cross-validation. After reconstructing all test folds, they are concatenated and the reconstruction quality score is calculated as the Pearson's r between the reconstructed and measured values. Features ranked by their reconstruction quality are shown for the drug response (left) and the CRISPR-Cas9 (right) datasets. Duplicated drug names represent replicated screens for the same drug. Representative examples of strongly selective CRISPR-Cas9 and drug responses are labeled. **b)** MOSA's partial dataset augmentation (missing value imputation) of drug IC50s compared to novel drug response screens. **c-e)**, similar to **b)**, using MOVE, MOFA and mean imputed values, respectively.

In summary, we believe the updated comparative analysis, including MOVE and mixOmics, alongside the original comparisons, provides a more comprehensive and rigorous evaluation of MOSA's capabilities in the context of multi-omics data imputation. We appreciate your guidance in enhancing the robustness of our study and hope that the revisions adequately address your concerns.

Methodological Soundness:

The study assumes data missing at random, which might not always be the case in omics studies, as some missing data could result from systematic biases. This assumption potentially limits the applicability of the learned VAE model, as it might not accurately impute data missing due to non-random factors.

This is a great point that has started to be addressed in recent studies^{28,29}. We agree with the reviewer that non-random patterns of data missingness can affect model training and generate potential confounding effects. These effects are often very difficult to detect and account for but we believe this is not strongly affecting our model.

In our datasets we have two types of missingness: 1) omics currently missing for a set of cell lines, i.e. data modality not measured in those samples; and 2) measurement missingness intrinsic to the data technology where features are not uniformly measured in all the samples, for example mass-spectrometry (MS) based datasets such as proteomics and metabolomics.

We cannot exclude that particular biological or technical aspects, such as cell lines that are hard to culture, can introduce biases in missing omics. Whilst we do not have any information supporting this we can observe, for example, in the drug response dataset that particular cell lines are more screened than others, hence creating non-random missing patterns (**Revision Figure 1**). We have further reduced this effect by only considering cell lines that have been characterized by at least two omics. However, this does not seem to be detrimental in the model's prediction capacity, where we observed strong correlation of reconstructed synthetic drug responses with an independent dataset (**Figure 2b**).

Revision Figure 1. Missing data pattern of drug response dataset. Dark blue represents missing drug response data.

Missing not at random (MNAR) can emerge based on intrinsic technical limitations of the assays, for example proteomics data where missingness is, in part, MNAR (Supplementary **Figure 5a**)³⁰. Our validation process involved correlating the proteomic profile of the cell lines with replicates from an independent dataset acquired with different MS technologies and in a different laboratory (**Figure 3a**). Through this, we were able to validate the model's capability in imputing missing values, specifically within the MNAR context. Our model is able to reconstruct cancer cell line proteomic profiles that approximate independent proteomic profiles, and notably even for cell lines for which no prior proteomics data is available (**Figure 3a**). This last point can only be achieved through associations identified using other orthogonal omics, e.g. transcriptomics where technology intrinsic missingness does not happen and the model can also robustly reconstruct the gene expression profiles (**Revision Figure 2**).

Revision Figure 2. Evaluation of MOSA on augmenting transcriptomic data. Samples that do not have any transcriptomic data were augmented with synthetic data that were highly correlated with external observed transcriptomic data for the same samples.

Taken together, these examples show that the model can generate robust synthetic data across different omics with different patterns of missing values, suggesting that this may not play a determinant role. One possible explanation for the robust performance of MOSA lies in its utilization of a large number of omic datasets, underpinned by a multi-view VAE architecture. Although the missingness pattern in some omics datasets may be MNAR, MOSA effectively leverages information from additional omic datasets to augment the data. This process is further enhanced by the inclusion of our view dropout technique, which prompts the model to learn to reconstruct missing data from other omic types in addition to its own. For example, if the entire proteomic view is masked as drop out during training, the model must rely on non-proteomic datasets for reconstructing the MNAR proteomic data. This strategy helps to prevent the model from learning a biased distribution, particularly when the observed data and missing data exhibit different distributions. Nevertheless, we agree it is important to further improve our models to more formally detect and model the missingness patterns in the data to potentially improve the model's reconstruction power. In the future work section of our paper, we included the following statement to address this:

"We also aim to address the complex challenge of data missing not at random (MNAR), a scenario commonly encountered in omics datasets, by adapting VAE architectures to more accurately identify and handle MNAR scenarios^{28,29,31}."

Reproducibility:

The study provides sufficient detail in its methods, facilitating reproducibility and further application in the field. The study also provided the imputed data.

References

1. Wekesa, J. S. & Kimwele, M. A review of multi-omics data integration through deep learning approaches for disease diagnosis, prognosis, and treatment. *Front. Genet.* **14**, 1199087 (2023).
2. Cai, Z., Poulos, R. C., Liu, J. & Zhong, Q. Machine learning for multi-omics data integration in cancer. *iScience* **25**, 103798 (2022).
3. Minoura, K., Abe, K., Nam, H., Nishikawa, H. & Shimamura, T. A mixture-of-experts deep generative model for integrated analysis of single-cell multiomics data. *Cell Rep*

- Methods* **1**, 100071 (2021).
4. Cohen Kalafut, N., Huang, X. & Wang, D. Joint variational autoencoders for multimodal imputation and embedding. *Nature Machine Intelligence* 1–12 (2023).
 5. He, Z. *et al.* Mosaic integration and knowledge transfer of single-cell multimodal data with MIDAS. *Nat. Biotechnol.* 1–12 (2024).
 6. Ghazanfar, S., Guibentif, C. & Marioni, J. C. Stabilized mosaic single-cell data integration using unshared features. *Nat. Biotechnol.* **42**, 284–292 (2024).
 7. Ashuach, T. *et al.* MultiVI: deep generative model for the integration of multimodal data. *Nat. Methods* **20**, 1222–1231 (2023).
 8. Hao, X. *et al.* MixGen: A New Multi-Modal Data Augmentation. *arXiv [cs.CV]* (2022).
 9. Liu, Z. *et al.* Learning Multimodal Data Augmentation in Feature Space. *arXiv [cs.LG]* (2022).
 10. Argelaguet, R. *et al.* MOFA+: a statistical framework for comprehensive integration of multi-modal single-cell data. *Genome Biol.* **21**, 111 (2020).
 11. Lotfollahi, M., Litinetskaya, A. & Theis, F. J. Multigrade: single-cell multi-omic data integration. *bioRxiv* 2022.03.16.484643 (2022) doi:10.1101/2022.03.16.484643.
 12. Du, J.-H., Cai, Z. & Roeder, K. Robust probabilistic modeling for single-cell multimodal mosaic integration and imputation via scVAEIT. *Proc. Natl. Acad. Sci. U. S. A.* **119**, 13e2214414119 (2022).
 13. Zhang, Z. *et al.* scMoMaT jointly performs single cell mosaic integration and multi-modal bio-marker detection. *Nat. Commun.* **14**, 384 (2023).
 14. Kircher, M. *et al.* Augmentation of Transcriptomic Data for Improved Classification of Patients with Respiratory Diseases of Viral Origin. *Int. J. Mol. Sci.* **23**, (2022).
 15. Partin, A. *et al.* Data augmentation and multimodal learning for predicting drug response in patient-derived xenografts from gene expressions and histology images. *Front. Med.* **10**, 1058919 (2023).
 16. Allesøe, R. L. *et al.* Discovery of drug-omics associations in type 2 diabetes with generative deep-learning models. *Nat. Biotechnol.* **41**, 399–408 (2023).

17. Rohart, F., Gautier, B., Singh, A. & Lê Cao, K.-A. mixOmics: An R package for 'omics feature selection and multiple data integration. *PLoS Comput. Biol.* **13**, e1005752 (2017).
18. Singh, A. *et al.* DIABLO: an integrative approach for identifying key molecular drivers from multi-omics assays. *Bioinformatics* **35**, 3055–3062 (2019).
19. Ewald, J. D. *et al.* Web-based multi-omics integration using the Analyst software suite. *Nat. Protoc.* (2024) doi:10.1038/s41596-023-00950-4.
20. Zhang, X., Xing, Y., Sun, K. & Guo, Y. OmiEmbed: A Unified Multi-Task Deep Learning Framework for Multi-Omics Data. *Cancers* **13**, (2021).
21. Mo, Q. *et al.* A fully Bayesian latent variable model for integrative clustering analysis of multi-type omics data. *Biostatistics* **19**, 71–86 (2018).
22. Mo, Q. *et al.* Pattern discovery and cancer gene identification in integrated cancer genomic data. *Proc. Natl. Acad. Sci. U. S. A.* **110**, 4245–4250 (2013).
23. Gong, B., Zhou, Y. & Purdom, E. Cobolt: integrative analysis of multimodal single-cell sequencing data. *Genome Biol.* **22**, 351 (2021).
24. Athreya, A. *et al.* Augmentation of Physician Assessments with Multi-Omics Enhances Predictability of Drug Response: A Case Study of Major Depressive Disorder. *IEEE Comput. Intell. Mag.* **13**, 20–31 (2018).
25. Ruepp, A. *et al.* CORUM: the comprehensive resource of mammalian protein complexes. *Nucleic Acids Res.* **36**, D646–50 (2008).
26. Szklarczyk, D. *et al.* The STRING database in 2017: quality-controlled protein-protein association networks, made broadly accessible. *Nucleic Acids Res.* **45**, D362–D368 (2017).
27. Chatr-Aryamontri, A. *et al.* The BioGRID interaction database: 2015 update. *Nucleic Acids Res.* **43**, D470–8 (2015).
28. Ipsen, N. B., Mattei, P.-A. & Frelsen, J. not-MIWAE: Deep Generative Modelling with Missing not at Random Data. *arXiv [stat.ML]* (2020).
29. Chen, J., Xu, Y., Wang, P. & Yang, Y. Deep Generative Imputation Model for Missing Not

At Random Data. in *Proceedings of the 32nd ACM International Conference on Information and Knowledge Management* 316–325 (Association for Computing Machinery, New York, NY, USA, 2023).

30. Poulos, R. C. *et al.* Strategies to enable large-scale proteomics for reproducible research. *Nat. Commun.* **11**, 3793 (2020).
31. Pereira, R. C., Santos, M. S., Rodrigues, P. P. & Abreu, P. H. Reviewing Autoencoders for Missing Data Imputation: Technical Trends, Applications and Outcomes. *jair* **69**, 1255–1285 (2020).

Point-by-point response to the reviewers' #3 comments

I agree with the previous reviewer that this study lacks systematic benchmarking by comparing with the existing tools. Experimental validations, which are completely missing, are also essential for the authors' claims. Therefore, I am afraid that I don't think the reviewers' concerns have been satisfactorily addressed. I am not convinced by the authors' claims that most of the existing tools are not feasible for comparison and that experimental validations are not needed.

The systematic comparison the reviewer proposes is more suited for a benchmark paper, not a method paper, as we propose here. As the reviewer mentions below, most existing methods do not handle 7 different types of omics data and, as we mention in detail in our revision table, these approaches have fundamental design and implementation limitations that hinder their adaptation to integrate all different omics available for cancer cell lines. Nevertheless, we successfully benchmarked our method against three broadly used, state-of-the-art approaches, showing positive results.

Considering the methodological focus of our paper, we consider it out of scope to perform extensive and complex biological follow-up validations. We propose to ameliorate this using an independent metabolomics dataset (PMID: 36321552) to validate our most novel biomarker associations. Furthermore, it is not uncommon to have methodological manuscripts published without tailored experimental validations, as these machine learning models are on their own trained over large datasets and they aim to help guide further follow-up studies that individually focus on a particular association which require extensive experiments to confirm the mechanism of the association.

If the paper is positioned as a method paper, MOSA has to be extensively tested and compared with ALL the existing tools currently in use. Of course, most of the existing methods were not designed to handle all 7 types of omics data, but that does not necessarily mean that they are not worth comparing with. Simply put, if MOSA is only usable for DepMap data, this study would not be suitable for publication as a method paper due to the limited scope.

As the reviewer acknowledges, the vast majority of current multi-omics approaches are not designed to handle all 7 types of omics considered here. This underscores the novelty and importance of our work.

We acknowledge that a systematic comparison would be important — albeit against all methods would arguably be unrealistic — but this would be a standalone benchmark paper, and not a method paper as we propose here.

Nonetheless, we have reviewed a total of 32 methods for multi-omics integration published since 2013. Some methods are designed for cross-modality translation tasks, where data from one modality is converted into a latent space using a specific encoder, and then used as input for a decoder from another data modality. Other methods need labelled data for supervised analysis and are tailored to specific omic modalities. Therefore, only 14 methods could be compared with the unsupervised multi-omics approach taken by MOSA. We have thoroughly tested these 14 methods, as described in the table provided in our point-by-point rebuttal and listed all the problems we faced. Many of the problems are associated with intrinsic design choices of the methods or the implementation that, as is, makes it impossible to apply them to the problem we set out to address. For example, StabMap expects datasets in different sets where features should be connected, trying to adapt our datasets to this has always rendered invalid formulations. Other examples, MIDAS, Multigrade, Cobolt, among others, are explicitly designed to handle count data assuming distributions (e.g., Poisson and Bernoulli), either by design or code implementation (e.g., requirement of integer-based variables), which do not support continuous datasets like those in the DepMap data. Of these, we have managed to run and benchmark against 3 independent methods, which MOSA has outperformed. To clarify our decision to benchmark against these three methods, we included Revision Table 1 as a Supplementary Table 10, and detailed our systematic search in the Methods section, containing detailed comparisons with other methods, even for those that could not be directly applied to our datasets and tasks.

We have conducted a thorough investigation of approaches applicable to the problem of cancer multi-omics integration. For the first time, we propose a model that successfully utilises all currently available large-scale datasets in cancer cell lines, adopting a holistic approach to identifying cancer dependencies. Furthermore, we disagree with the reviewer's suggestion for extensive formulation and implementation changes. We argue that if such extensive changes are

necessary for a method to be usable, their comparison becomes either uninformative or simply impossible.

If the paper is positioned as an analysis type of article, the authors would need to present the new insights and novel findings from their study. In addition, the analysis results should be presented in a more user-friendly way, for example, a web server that allows user to freely explore the imputation and original data and mine for biological insights based on their interests, like what DepMap has done.

Nevertheless, certain levels of experimental validations of the key discoveries from the pipeline are surely needed to prove that the analysis pipeline of MOSA makes sense in biology and is of unique value for the community.

Our study presents the most comprehensive machine learning model for cancer cell lines to date, used to conduct an extensive multi-omics biomarker discovery analysis. This analysis poses significant computational challenges, particularly due to the unprecedented scale required to apply feature explainability approaches to large numbers of features. Through this rigorous process, we have identified a previously underappreciated metabolomics biomarker — a dataset often missing from multi-omics integration and biomarker discovery pipelines. Particularly, our approach has highlighted the association of 1-methylnicotinamide with drug resistance, which is supported by recent single-cell drug resistance experiments from Aviv Regev's lab (PMID:34381210). These experiments demonstrated increased 1-methylnicotinamide abundance in cancer cells resistant to targeted therapies. We agree with the reviewer that further experimental work, such as inducing metabolite depletion and restoration to measure the response to drug treatment, would provide deeper mechanistic insights into this association. However, we argue that such experiments are time-consuming, complex, and beyond the scope of a methodology paper. Nonetheless, we have integrated a recent metabolomics dataset from 180 cancer cell lines (PMID: 36321551) in this revised manuscript to validate this metabolic drug resistance biomarker (new Supplementary Figure 8).

As recommended by the reviewer, upon publication, we will enhance the accessibility of the synthetic datasets by making them available for download directly from the CellModelPassports data portal (cellmodelpassports.sanger.ac.uk), including a dedicated download section for the synthetic datasets generated by MOSA. Additionally, we are developing a prototype that

integrates the synthetic measurements with the rest of the omics data, allowing users to easily navigate, visualise and toggle on & off these measurements.

Below we include a point-by-point response to the reviewer's comments and highlighted in blue the manuscript changes.

Point-by-point response to the reviewers' #3 comments

1. I think that the authors should include a comprehensive evaluation of the existing methods in the main text. In other words, information in supplementary table 10 should be presented in the main text.

As suggested by the reviewer, we now discuss in the main text our comprehensive evaluation of existing methods for multi-omics integration, describing the different types of methods, outlining limitations, and explaining our rationale for selecting the three main methods used in our benchmarks: MOFA, MOVE, and mixOmics. This table is also re-ordered from Supplementary Table 10 to Supplementary Table 6, as a result of being included in the main text.

2. I am still not convinced that MOSA can only be compared with other methods that can handle all the 7 omics data. It should be done, and can be done, to compare the data augmentation results of MOSA (with multi-omics data) with the results of other methods (for example JAMIE, which only used 2 omics). Again, the authors need to explicitly show that MOSA outperforms the others even if they only handles smaller datasets.

We agree that comparisons with methods that integrate fewer methods is possible, albeit with a fundamental limitation: it becomes impossible to synthetically reconstruct an omic that is not considered in the model. Furthermore, some methods require samples to be matched across omics, meaning they rely on the intersection of samples. In contrast, MOSA can reconstruct omics that are completely absent in certain samples by leveraging information from the other orthogonal omics available for those samples. Therefore, restricting the analysis to fewer omic types not only limits the model's capacity but also necessitates the selection of specific omics for integration, which can be a non-trivial task.

Following the reviewer's comment, we benchmarked MOSA with other methods that integrate 2 omics, including JAMIE¹ as suggested. For data integration we chose transcriptomics and drug response as they are important for the clustering and synthetic data generation benchmarks designed in this study. Moreover, many of the methods we evaluated were designed for

transcriptomics data. From the list of evaluated methods, in addition to JAMIE¹, we choose scVAEIT², iClusterPlus³, and moCluster⁴ which had particularly good performance in previous benchmarks⁵ and were complementary with the other methods already selected for the benchmark. These four methods were particularly suitable for our data, as other methods require specific input formats, such as count data from RNA-seq or peak data from ATAC-seq, or they rely on distributions that are not appropriate for our dataset. By selecting these methods, we ensured compatibility with our data and enhanced the diversity of approaches included in the benchmark. We have now integrated four new methods into our analysis, bringing the total to seven independent methods in our benchmark comparison.

Supplementary Figure 6. MOSA benchmark against two omics integration. **a)** correlation of reconstructed transcriptomics against an independently processed transcriptomics dataset. **b)** cancer cell line correlations (Pearson's r) between an independent drug response dataset (CTD2^{6,7}) and the MOSA reconstructed dataset, grouped by whether the cancer cell line had prior availability of drug response in the datasets for the model training versus cell lines without drug response data. **c)** tissue-type comparison of cell line separations quantified by Calinski-Harabasz index (higher value indicates better) and Davies-Bouldin index (lower value indicates better). For better visualization, scVAEIT is not included in this comparison due to its suboptimal performance.

MOSA consistently outperformed the other methods when trained on either two or seven omics, and we have incorporated these results in the manuscript (**Supplementary Figure 6**). Specifically, when using two omics, MOSA demonstrated better performance in imputing partially missing drug response data (**Rebuttal Figure 1**). Moreover, the integration of seven omics resulted in a significant improvement (p -value < 0.0001) over two omics in the synthetic

reconstruction of gene expression, as evaluated using an independent dataset (**Supplementary Figure 6a**). For the reconstruction of drug response data (**Supplementary Figure 6b**), MOSA trained with seven omics also achieved the best performance. While the difference between MOSA with seven omics and MOSA with two omics in drug response reconstruction was less pronounced, the seven-omics configuration consistently outperformed other methods across both tasks. Of note, mixOmics, iClusterPlus and moCluster do not have the capability to generate synthetic data, a common limitation of current approaches and a particular innovation of our method.

For tissue-type clustering, MOSA produced more distinct and well-defined clusters (**Supplementary Figure 6c and Rebuttal Figure 2**). Interestingly, using only transcriptomics and drug response data resulted in better tissue-type clustering compared to the full model with seven omics. This can be explained by the fact that transcriptomics and drug response are highly tissue-specific, whereas other omics, such as metabolomics and proteomics, are only partially structured by tissue and capture other factors more strongly, such as the epithelial–mesenchymal transition phenotypes^{8–11}. Therefore, when including omics that are more aligned with tissue-type, clustering by tissue will naturally be better.

We thank the reviewer, for the constructive suggestion, which we believe has strengthened our manuscript. We have incorporated this analysis into the main text. Our results demonstrate that MOSA outperforms methods integrating fewer omics, and that increasing the number of omics further enhances the data augmentation process, underscoring the importance of having methods tailored to integrate multi-omics datasets. Collectively, we are confident that these additions comprehensively address the reviewer’s comment.

Rebuttal Figure 1. Partial dataset augmentation (missing value imputation) of drug IC50s compared to novel drug response screens, using transcriptomics and drug response, for all benchmarked methods **a-f** for MOSA, MOFA, MOVE and JAMIE, scVAEIT and the mean, respectively.

Rebuttal Figure 2. Latent space visualization comparison using transcriptomics and drug response across all benchmarked methods. **a-h)** UMAP representation of the trained latent dimensions, colored according to tissue of origin. Color palette is the same as Figure 1c. **i)** comparison of cell line tissue type separations quantified by Calinski-Harabasz index (higher value indicates better) and Davies-Bouldin index (lower value indicates better). For better visualization, scVAEIT is not included in this comparison due to its suboptimal performance.

References

1. Cohen Kalafut, N., Huang, X. & Wang, D. Joint variational autoencoders for multimodal imputation and embedding. *Nat. Mach. Intell.* **5**, 631–642 (2023).
2. Du, J.-H., Cai, Z. & Roeder, K. Robust probabilistic modeling for single-cell multimodal mosaic integration and imputation via scVAEIT. *Proc. Natl. Acad. Sci. U. S. A.* **119**, e2214414119 (2022).
3. Mo, Q. *et al.* A fully Bayesian latent variable model for integrative clustering analysis of multi-type omics data. *Biostatistics* **19**, 71–86 (2018).
4. Meng, C., Helm, D., Frejno, M. & Kuster, B. MoCluster: Identifying joint patterns across multiple omics data sets. *J. Proteome Res.* **15**, 755–765 (2016).
5. Cai, Z., Poulos, R. C., Liu, J. & Zhong, Q. Machine learning for multi-omics data integration in cancer. *iScience* **25**, 103798 (2022).
6. Seashore-Ludlow, B. *et al.* Harnessing Connectivity in a Large-Scale Small-Molecule Sensitivity Dataset. *Cancer Discov.* **5**, 1210–1223 (2015).
7. Rees, M. G. *et al.* Correlating chemical sensitivity and basal gene expression reveals mechanism of action. *Nat. Chem. Biol.* **12**, 109–116 (2016).
8. Menden, M. P. *et al.* Machine learning prediction of cancer cell sensitivity to drugs based on genomic and chemical properties. *PLoS One* **8**, e61318 (2013).
9. Gonçalves, E. *et al.* Pan-cancer proteomic map of 949 human cell lines. *Cancer Cell* **40**, 835–849.e8 (2022).
10. Shorthouse, D., Bradley, J., Critchlow, S. E., Bendtsen, C. & Hall, B. A. Heterogeneity of the cancer cell line metabolic landscape. *Mol. Syst. Biol.* **18**, e11006 (2022).
11. Cherkaoui, S. *et al.* A functional analysis of 180 cancer cell lines reveals conserved intrinsic metabolic programs. *Mol. Syst. Biol.* **18**, e11033 (2022).